# SnowQM 1.0: A fast *R* Package for bias-correcting spatial fields of snow water equivalent using quantile mapping

Adrien Michel[1,2], Johannes Aschauer[1], Tobias Jonas[1], Stefanie Gubler[2], Sven Kotlarski[2], and Christoph Marty[1]

[1]WSL Institute for Snow and Avalanche Research SLF, Davos, Switzerland
[2]Federal Office of Meteorology and Climatology MeteoSwiss, Zurich, Switzerland
**Correspondence:** Adrien Michel (adrien.michel@meteoswiss.ch)

**Abstract.** Snow plays a crucial role in regional climate systems worldwide. It is a key variable in the context of climate change because of its direct feedback to the climate system, while at the same time being very sensitive to climate change. Long-term spatial data on snow cover and snow water equivalent are scarce, due to the lack of satellite data or forcing data to run land surface models back in time. This study presents an *R* package, SnowQM, designed to correct for the bias in long-term spatial snow water equivalent data compared to a shorter-term and more accurate dataset, using the more accurate data to calibrate the correction. The bias-correction is based on the widely applied quantile mapping approach. A new method of spatial and temporal grouping of the data points is used to calculate the quantile distributions for each pixel. The main functions of the package are written in *C*++ to achieve high performance. Parallel computing is implemented in the *C*++ part of the code.In a case study over Switzerland, where a 60-year snow water equivalent climatology is produced at a resolution of 1 day and 1 km, SnowQM reduces the bias in snow water equivalent from -9 mm to -2 mm in winter and from -41 mm to -2 mm in spring. We show that the *C*++ implementation notably outperforms simple *R* implementation. The limitations of the quantile mapping approach for snow, such as snow creation, are discussed. The proposed spatial data grouping improves the correction in homogeneous terrain, which opens the way for further use with other variables.

## 1 Introduction

Snow is a central component of the climate system in many regions worldwide. It influences the local energy balance, air temperature, and wind (Barry, 1996; Serreze et al., 1992). Snow cover and snow season duration impact permafrost, soil biology (Smith et al., 2022) and vegetation (Rumpf et al., 2022). Snow is also a crucial component of the hydrological cycle acting as a buffer for precipitation. Snow accumulation and melt will directly influence discharge and temperature in mountainous catchments (Michel et al., 2022), impacting water availability, energy production and ecosystems downstream (Schaefli et al., 2007; Beniston, 2012).

In the context of climate change, snow cover plays a key role by having a direct feedback on the climate system through changes in surface albedo and surface temperature, and being at the same time very sensitive to changes in near-surface air temperate and precipitation (Pörtner et al., 2019). Despite this importance of snow, long-term records of the spatial distribution of snow height or snow water equivalent – e.g. based on satellite data, reanalysis products, or spatially interpolated station series – exist only at rather low spatial resolution and the available products are poorly matched to each other (Terzago et al., 2017; Luojus et al., 2021). The *R* package (R Core Team, 2021) presented here, called SnowQM, was developed to provide a homogeneous long-term gridded data set of snow water equivalent for Switzerland between 1962 and 2021. This is achieved by applying quantile mapping to correct the bias in the snow water equivalent as simulated by a simplified model of the surface snowpack over the period 1962–2021, using a more accurate simulation benefiting from in-situ data assimilation available over the period 1999–2021 as calibration reference.

Quantile mapping is a widely used approach in climatology to correct climate model output relative to observations (Ivanov and Kotlarski, 2017; Holthuijzen et al., 2022), to correct long-term data series of lower quality with the help of shorter, higher quality datasets (Rabiei and Haberlandt, 2015), and to spatially transfer meteorological time series (Rajczak et al., 2016; Michel et al., 2021). Quantile mapping has also already been used to correct snow water equivalent maps (Jörg-Hess et al., 2014) and snow cover fraction projections (Matiu and Hanzer, 2022). Jörg-Hess et al. (2014) demonstrate the usability of quantile mapping to correct spatialised snow datasets, highlighting the problem of the binary behaviour of snow (snow vs no snow) and the difficulty of quantile mapping to remove bias in this respect (and not create additional bias). This issue is also discussed in the current paper. Matiu and Hanzer (2022) show that when applying quantile mapping to the snow cover fraction, not using a moving window approach to calculate the quantile distribution leads to spurious breaks in the data. In SnowQM, quantile mapping is chosen over the more recent machine learning based approach (e.g. King et al., 2020) because of its simplicity and relatively low computational training cost compared to machine learning.

When quantile mapping is applied separately to each pixel of a grid as is commonly done (see e.g. CH2018, 2018), the spatial structure of the data may be poorly matched to that of the observations. There are several approaches that try to overcome this limitation, many of them based on the shuffling and reordering of the data introduced by Clark et al. (2004). In SnowQM, we propose a parameterisation for the construction of the quantile distributions by temporally and/or spatially grouping the data for each pixel and evaluate how it helps to preserve the spatial dependence of the data, which to our knowledge is not present in the literature.

The SnowQM package is distributed with a built-in toolkit to evaluate the quality of the correction performed. In addition, although tailored towards snow water equivalent to account for some specifics of this variable, the computationally efficient SnowQM kernel can easily be used to any other gridded data set. Finally, SnowQM is distributed as a user-friendly *R* library, but the core of the model is coded in *C++* to achieve significantly higher serial and parallel computational performance than *R*. This is done without any additional complication for the user as *R* automatically compiles the *C++* code in the background at installation time (as with many standard *R* libraries).

The development of SnowQM is part of a joint project between the Swiss Federal Office of Meteorology and Climatology, MeteoSwiss, and the WSL Institute for Snow and Avalanche Research, SLF. The aim of the project is to obtain a long-term

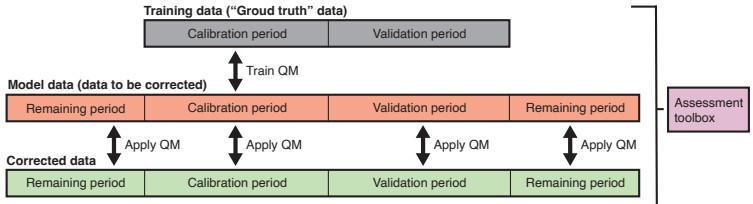

**Figure 1.** Conceptual view of the different data used and of the main steps of SnowQM. Note that calibration and validation periods do not necessarily have to be continuous in time but could also be sampled intermittently from the entire training period.

climatology of snow water equivalent (for research purposes) and snow height (for public purposes) that is operationally updated on a daily basis. In this project, SnowQM is used to produce the snow water equivalent data and the model SWE2HS (Aschauer et al., 2023) is used to convert the snow water equivalent to snow height. The operational use of SnowQM adds
some constraints to the development (e.g. working with daily data rather than already temporally aggregated data, although climatological analysis is not performed on a daily timescale). The full model chain has been tested during winter 2022-2023, will be pre-operational internally during winter 2023-2024, and the automatically generated operational analysis (plots) will be publicly available during winter 2024-2025. The full publicly available dataset will be updated at the end of each winter season.
In this paper we first present, in detail, SnowQM principles and its implementation along with the assessment toolbox. Then, an example of an application based on two different snow model outputs from Switzerland is presented to demonstrate how SnowQM can be used to produce a homogeneous timeseries of snow water equivalent maps over a long time period. Based on this example, the robustness and limitations of SnowQM are assessed and advice for further usage is provided. In addition, thanks to the availability of many SnowQM runs with different setups, some interesting insight in the quantile mapping method
in general is provided. The snow climatology produced for Switzerland is used here as an example, and will be studied and validated in more detail in a future work. The package comes with a Vignette showing a step-by-step example and giving information of how to use the package for other applications, e.g. on other variables or for spatial downscaling.

## 2    Model description

SnowQM is an *R* library for correcting snow water equivalent (SWE) grids, called the model data, to match another set of SWE
grids considered as ground truth and called the training data. The SWE correction is performed by QM (quantile mapping). The quantile distributions construction (Section 2.2.1) depends on 7 free parameters (Section 2.2.2). The quantile distributions are calculated over a calibration period during the training phase of the QM and are applied to both the calibration and an independent validation period to assess the correction quality. Once trained, the QM can also be applied to the remaining part of the model data not used for calibration and validation (see Figure 1).

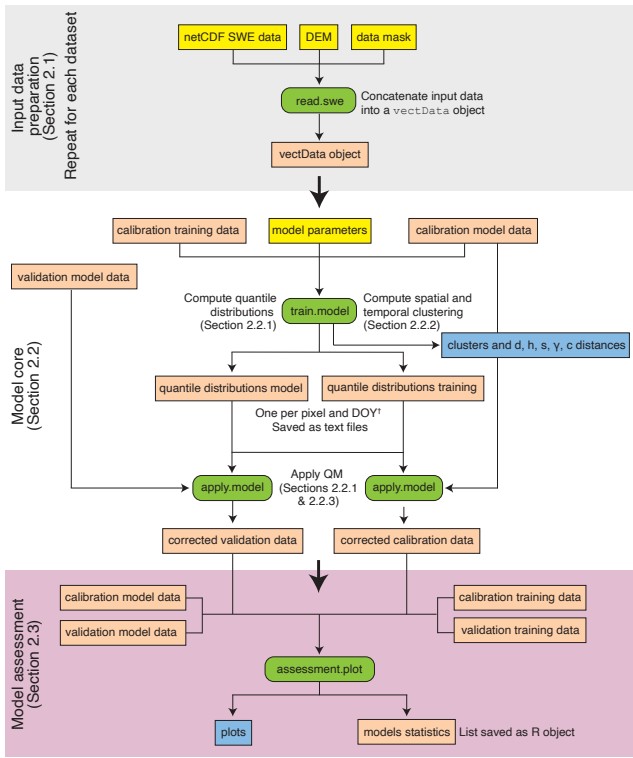

**Figure 2.** Main workflow of SnowQM. The yellow labels represent the inputs to SnowQM, the green labels the main internal *R* functions, the orange labels the data created, and the blue labels the PDF graphics produced. The different parts of the figure are detailed in the sections indicated.

† DOY means day of the year.

Tools built into the assessment toolbox allow the quality of the correction to be assessed using 14 different metrics (section 3). Results are saved for later analysis. The evaluation tools also create different maps to visually assess the performance of the the QM (saved as pdf files). Evaluation can be performed for different regions separately if region masks are provided. The library workflow and the most important functions are shown in Figure 2.

The library is structured around three main parts: the I/O (section 2.1, gray part of Figure 2); the core QM (section 2.2, white part of Figure 2); and the performance evaluation tools (section 2.3, purple part of Figure 2). The *R* library is provided with examples and detailed documentation.

## 2.1 I/O and data format

Model and training data have to be provided at daily resolution as netCDF files spanning the same time periods. To calibrate the free parameters, model and training data should be split between a calibration and a validation period (which are not necessarily continuous in time). Figure 1 illustrates the nomenclature used for the different datasets in SnowQM. In order to use the internal

algorithm for the spatial grouping of the data (see Section 2.2.2), a digital elevation model (DEM) should be provided. The slope, the aspect angle ($\gamma$, i.e. north-south exposition, in degree, with $0°$ being the north), and the curvature are computed from the DEM using the implementation of same algorithm as in the MeteoIO library (Bavay and Egger, 2014). Exact details about the input, such as the required dimension and variable names in the netCDF files, can be found in the package documentation.

Inputs grids are internally stored in *R*-objects called `vectData`, which are the core objects of SnowQM. These objects allow for easy manipulation and masking, lower memory usage, fast operations on the spatial or temporal dimensions, and light interface with *C++*. SnowQM is provided with an overloaded plot function to obtain maps from these objects and with I/O functions to convert netCDF to `vectData` objects and vice versa. `vectData` objects can easily be masked using ASCII or GeoTIFF grids as mask using the provided input and masking functions. The DEM should be provided as ASCII or GeoTIFF

grids. Full details about the `vectData` object are given in the Appendix A.

     The quantile distributions computed during the training phase of the model (Section 2.2.1) are written as text files. One text file is written per pixel (using the pixel coordinates as file name), containing the quantile distributions for each day of the year (DOY). The corrected grids are written as `vectData` objects (which should be provided as input for the assessment tools) and converted into netCDF files.

## 2.2   SnowQM core

### 2.2.1   Quantile mapping

QM is a distribution-based bias correction approach. Its principle is to correct the model data ($SWE_{mo}$) so that these approximate the distribution of the training data ($SWE_{tr}$) using the following formula:

$$SWE_{co} = F_{tr}^{-1}(F_{mo}(SWE_{mo})), \tag{1}$$

where $SWE_{co}$ is the corrected dataset, $F_{mo}$ is the cumulative distribution function (CDF) of $SWE_{mo}$, and $F_{tr}^{-1}$ is the inverse of the CDF of $SWE_{tr}$.

     There are a number of different approaches to approximate $F_{mo}$ and $F_{tr}^{-1}$, which can be either empirical or parametric (see, among others, Maraun, 2016). Based on the work of Gudmundsson et al. (2012), the functions are approximated using the empirical quantile distribution, i.e. the data are sorted in ascending order and simply separated into $n-1$ bins of equal

number of points, where $n$ is the number of quantiles used. This approach has already been widely used in climatology (see, for example, Themeßl et al., 2012; Wilcke et al., 2013; Rajczak et al., 2016). Several different numerical implementations exist to compute the empirical quantile distributions and the exact position of the quantiles (Hyndman and Fan, 1996). Indeed, once the sample has been partitioned into $n-1$ bins, the question of where exactly to take the quantile value between the max value of a given bin and the min value of the next bin is subjective. Our implementation corresponds to the 7[th] definition in Hyndman

and Fan (1996), from Gumbel (1930), which is a linear interpolation between the two closest values using the quantile position. In this work, 101 quantiles are used (the quantiles are numbered between 0 and 100), which means 100 bins.

     To calculate the correction for a specific value in the model data, the SWE value to be corrected is compared to the quantile distribution of $SWE_{mo}$ to see in which quantile it lies. Here, no interpolation between quantiles as in Rajczak et al. (2016)

is used, the quantile used is the first quantile less than or equal to the SWE value under consideration. Then, the absolute difference between this particular quantile of $SWE_{mo}$ and the corresponding quantile of $SWE_{tr}$ is applied as correction. As in the work of Themeßl et al. (2012) and Rajczak et al. (2016), the extreme values belonging to the 100th quantiles (i.e. $\geq \max(F_{mo})$) are corrected according to the 99th quantile. The QM is applied to each pixel and each DOY separately.

### 2.2.2 Temporal and spatial grouping - Free parameters

In order to appropriately capture the different seasonal signals, the QM correction function is computed separately for each DOY as in the works of Thrasher et al. (2012) and Rajczak et al. (2016), which is not always done in QM applications (see e.g. Grillakis et al., 2017; Cannon, 2018). Meaningful CDF and quantile distributions cannot be calculated by simply using each DOY and pixel separately because there are usually not enough data points available (unless several centuries of data are available). It is therefore necessary to group the data along spatial or temporal dimensions. Many QM applications only use temporal grouping (see e.g. Jörg-Hess et al., 2014; Rajczak et al., 2016). SnowQM can handle spatial and temporal grouping simultaneously or separately. Temporal grouping is straightforward: given a time window parameter $w_t$, the CDF for DOY $i$ is calculated using all SWE values for the pixel of interest in the time interval $i \pm w_t$ of each year. Spatial grouping is more difficult. A simple approach would be to take all points within a given radius. However, to work with variables influenced by the topography, a grouping of pixels according to topographic similarities (e.g. a minimum difference in elevation) is advisable. Taking all pixels that are similar according to certain parameters (e.g. all pixels in a given radius with a given maximum elevation difference) leads to larger groups of data for some pixels (e.g. in flat terrain) and smaller ones in other areas (e.g. high relief terrain). The consequence is an unbalanced correction between pixels and an increase in computational time for pixels with a large grouping (see section 3.4.3).

In our implementation, the similarity between pixels is computed in the 5D space $d-h-s-\gamma-c$, where $d$ is the distance to the pixel of interest in the horizontal plane, $h$ is the difference in elevation, $s$ the difference in slope, $\gamma$ the difference in aspect, and $c$ the difference in curvature. Given the 5 parameters $\Delta d$, $\Delta h$, $\Delta s$, $\Delta \gamma$, $\Delta c$, and an additional parameter $\Sigma P$, which is the minimum number of pixels to be considered, an iterative approach is used to compute the spatial grouping of each pixel separately. A hyperrectangle is constructed around each pixel in the $d-h-s-\gamma-c$ space. At each iteration, the length of the edges of the hyperrectangle is increased by $\Delta d$, $2\Delta h$, $2\Delta s$, $2\Delta \gamma$, and $2\Delta c$ in the respective dimensions until at least $\Sigma P$ neighbouring pixels are included (the factor of 2 in some dimensions is due to the fact that the size is increased in both the positive and negative directions). Upper boundaries in edge size are used for each dimension to avoid infinite iterations if $\Sigma P$ is not reachable, and to avoid grouping with too distant pixels (100 km in the $d$-dimension, $\pm$ 400 m in the $h$-dimension, $\pm$ 90° in the $s$-dimension, $\pm$ 180° in the $\gamma$-dimension, and $\pm$ 0.1 m$^{-1}$ in the $c$-dimension). These values can be changed by the user (in the source code).

SnowQM therefore has 7 free parameters summarised in table 1. The physical interpretation of $\Delta d$, $\Delta h$, $\Delta s$, $\Delta \gamma$, and $\Delta c$ is as follows: for small values, the model will use data from neighbouring pixels that are close in the given dimension, while for large values, pixels that are very far apart in the respective dimension can also be chosen. Using the built-in function `train.model` (see Figure 2), maps indicating for each pixel the number of pixels grouped in space (which may be less than

$\Sigma P$ if the limits are reached, or greater since iteration stops when at least $\Sigma P$ pixels are included), and the final value of $d$, $h$, $s$, $\gamma$, and $c$ are produced. The smallest group size among all pixels is printed to check that a sufficiently large number of points are

used to calculate a meaningful CDF distribution. Note that this approach differs from traditional clustering as in Gutiérrez et al. (2004). SnowQM does not create clusters of similar pixels to gain computational efficiency by performing a single calculation for the entire group. Instead, it takes data from a group of similar pixels to construct a more representative quantile distribution for the pixel of interest, but each pixel uses its own group.

**Table 1.** Summary of the free parameters.

| Name | Description | Name of the parameter in the library |
|------|-------------|--------------------------------------|
| $w_t$ | size of the time window (applied in positive and negative direction) | `temporal.step` |
| $\Sigma P$ | minimum number of pixel to be included in the spatial grouping | `target` |
| $\Delta d$ | step in the spatial direction (radius) | `spatial.step` |
| $\Delta h$ | step in the elevation direction (applied in positive and negative direction) | `elevation.step` |
| $\Delta s$ | step in the slope angle direction (applied in positive and negative direction) | `slope.step` |
| $\Delta \gamma$ | step in the aspect angle direction (applied in positive and negative direction) | `aspect.step` |
| $\Delta c$ | step in the curvature direction (applied in positive and negative direction) | `curvature.step` |

### 2.2.3 The zero quantiles problem

Using QM on continuous values, such as air temperature, is straightforward. When used on data with many zeros, such as precipitation or SWE, the application is more difficult, because in this case many quantiles will have a zero value. When a zero value in the original model data has to be corrected, there are thus many quantiles to which it can be assigned. If the CDF of the training data has more zeros, this is not a problem because any zero will be mapped to a zero and hence no correction will be applied. But otherwise, there is no way to decide whether a zero should be mapped to the zero part of the training CDF (i.e.

no correction), or to the positive part of the CDF (i.e. snow should be created). This situation is illustrated in Figure 3. In the case of precipitation, a probabilistic frequency adaptation approach can be used by randomly choosing one of the quantiles of the training distribution corresponding to a zero value in the model distribution. On a long-term scale (i.e. enough data points re considered), this leads to a precipitation frequency and accumulated mass similar to the training data (Rajczak et al., 2016). In the case of snow, such an approach is not possible. In fact, the SWE value depends not only on the accumulation at the

current time step, but also on the value at the previous time step. Consequently, a probabilistic approach cannot be used to decide whether snow should be present at a given time step. This issue has already been highlighted in Jörg-Hess et al. (2014).

The implementation chosen here is rather conservative. When a zero value is encountered in the model data, the library chooses between two options: 1) If there is snow in the previous time step of the corrected time series, the SWE is divided by 2, i.e. we have an exponential decay instead of a sudden jump to no snow due to a zero value (the minimum SWE is set

to 0.5 mm); 2) If the previous timestep had no snow, the SWE is corrected using the lowest quantile (which, except for pixels

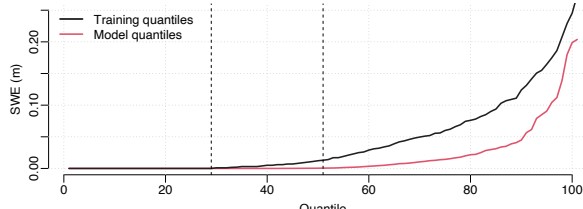

**Figure 3.** Quantile distribution for the training data (black) and the model data (red) for the 1st of April for a pixel located at coordinate $chx = 575'000$ and $chy = 100'000$ (coordinate system CH1903 / LV03 (EPSG:21781), see also Figure 4) computed with $w_t = 30$ and $\Sigma P$ =1 over the odd years of the period 1999–2021. Between the two dashed lines, the training distribution is greater than zero, while the model distribution is zero.

that contain snow all year round in the training data, will lead to a SWE value remaining at zero). With this approach, we almost never allow the QM correction to create snow when the model data contains no snow. As a result, SnowQM cannot fully correct pixels for which the start of the snow season occurs later in the model data than in the training data. However, without additional information, it is not possible to decide whether snow should be created or not.

## 2.3 Metrics and assessment tools

SnowQM provides a built-in evaluation toolkit to assess the quality of the QM procedure. To do this, the training data is compared to the model data on one side and to the corrected data on the other side. The error of the model data (i.e. the difference to the training data) can then be compared with the error of the corrected data to evaluate the benefits of the QM correction. This procedure can be applied if either all model data is used for training, or if the model data is split into calibration and validation data (in this case, errors are calculated separately for the calibration and validation periods). The evaluation toolkit produces figures and maps explained below, and saves the spatially averaged values of the metrics as a *R* `list` wrapped in a `RDS` file (*R* data format).

The evaluation toolkit computes 14 metrics, separated into 4 categories detailed below, the aim being to reduce the 3D information in the data to a few numbers to facilitate comparison of the parameters and to calibrate them. The metrics cover different aspects of the errors, and users should use a subset of the metrics relevant to their application. In order to be more versatile, it is possible to provide a set of region masks to the function and all metrics and graphs will be produced separately for each region based on a single run output. The evaluation toolkit is implemented independently of the core QM routines, which means that it can be used separately to compare any SWE grid product and benefit from the fast implementation in *C++* wrapped in a user-friendly *R* interface.

For the analysis, meteorological seasons are used, they are defined as DJF (December, January, February), MAM (March, April, May), JJA (June, July, August), and SON (September, October, November). Also, throughout the whole manuscript, the year $n$ corresponds to September $n-1$ to August $n$. The 14 metrics are calculated as follows:

1. The average seasonal error (8 values: ME SWE and MAE SWE for each season). The average SWE difference (bias) between datasets is calculated for each pixel and each season of each year separately. The average of the values for all pixels and all years is calculated and saved, giving a mean error (ME, or bias) value per season. The mean absolute error (MAE) is calculated in the same way except that negative bias values are multiplied by -1 before averaging. The ME and MAE values are plotted as a boxplot per season. All years are averaged together for each pixel, to have one map per season representing the average error of each pixel (see for example Figure 6).

2. The duration of the snow season (2 values: ME and MAE season length). For each pixel, the duration of the snow season is calculated. It is defined as the longest series of consecutive snow days between September and August. Snow days are days where the SWE is above a given SWE threshold value (parameter to be provided by the user, in all examples of this article we use 5 mm SWE). Before the calculation, the SWE time series is smoothed with a moving window average (the window size is also provided by the user, here 7 days are used). For each year, the relative bias of the snow season duration with respect to the training dataset is calculated. As with the seasonal bias, the ME and MAE are calculated over all pixels and all years together, and the ME for each pixel (averaged over all years together) is plotted on maps.

3. False negative and false positive rates (2 values). For each pixel and timestep, a confusion matrix is calculated based on the snow day status (i.e. SWE greater than a threshold, the same used in the snow season computation). The timestep can be a true positive, a true negative, a false positive, or a false negative. The total ratio of each of the four states is plotted as a barplot. The different ratios of each state are also plotted for each pixel in maps. The same graphs are produced for all seasons separately. As final measures, the ratios of false positive and false negatives are used.

4. The spatial SWE pattern (2 values: ME and MAE of spatial pattern). In order to asses to what extent the QM modifies the spatial pattern of SWE, a spatial metric is defined as follows: for a given radius (user provided parameter, here 5 km is used), the average SWE of the pixel of interest over the entire investigated period is divided by the average SWE of the surrounding pixels. This gives a map of the spatial local anomalies compared to the surrounding. Subsequently, the ME and MAE are calculated by comparing the spatial pattern maps for the two datasets. Proper statistical correlation with neighbouring pixels is not used because this method is not adapted for time series with many zeros.

An example of the use of the metrics is given in section 3.2.

## 2.4 *C++ implementation*

SnowQM is mainly written in *R* with the core functions written in *C++-11* using the *Rcpp* package (Eddelbuettel and Balamuta, 2018). The *C++* code is parallelized using *OpenMP* (OpenMP Architecture Review Board, 2021) to achieve high performance while hiding most of the complexity from the user. SnowQM also depends on the following *R* libraries: *lubridate* (Grolemund and Wickham, 2011), *foreach* (Microsoft and Weston, 2022), *ncdf4* (Pierce, 2021), *ncdf4.helpers* (Bronaugh, 2021), *fields* (Douglas Nychka et al., 2021), and *raster* (Hijmans, 2023).

All the code was first written and tested in *R* to show the suitability of the method. Profiling was then carried out using gperftools (gperftools, 2022) to identify bottlenecks to be implemented in *C++* (this language was used for simplicity as it is easy to interface with R and known to the main developer, but other languages such as *C*, *Fortran* or *Python* could achieve similar performance). The main bottleneck was identified in the evaluation module, where *R* was slow to compute aggregated values across multiple dimensions of large datasets. Porting to *C++* allowed to optimise the way the data is stored in memory for each different metric, allowing a fast computation. Note that although *Rcpp* provides direct access to objects that store the *R* variable without any copy, these objects are not thread-safe. In order to implement parallel computation (see below), a copy to a standard *c++* container is required anyway, allowing the possibility to optimise the data organisation. For the training and correction phase of the model, the part where the proper quantile computation is performed is not problematic (*R* has an already well optimised `quantile` function). The slower part is also the data access and subsetting, which can be accelerated using *C++* and again an optimal data structure.

The second step of optimisation was the implementation of parallel computing. *R* offers some parallel possibilities. But *R* was not designed for parallel computing, and the underlying *C* structures used to store data are not thread-safe. Thus, parallel regions in *R* imply a full copy of memory for each thread, which is a major limitation when using data that occupies several gigabytes of RAM. We took advantage of the fact that the main core functions were already implemented in *C++* to parallelise the code directly in the *C++* functions using *OpenMP*.

The performance improvements are shown in section 3.4.3. We acknowledge that these performance gains come at the cost of more complicated code, especially for environmental scientists who are usually more proeficient with *R* than *C++*. However, using *Rstudio* allows users to hide all the complications of *C++* compilation. In addition the *C* or *C++* interfaces are really common in many *R* packages. Apart from the *OpenMP* library, which now comes with every *C++* compiler on Windows and Linux (and is readily available on MacOS), no other non-standard *C++* library is used, which makes installation easier. SnowQM has been tested on Linux, MacOS and Windows, and on a wide range of hardware from laptops to high performance computing (HPC) installations such as the Swiss Centre for Super Computing (CSCS).

## 3 Example application over Switzerland

As a test and example, we apply SnowQM to produce a daily SWE dataset between 1962 and 2021 covering the area of Switzerland (around $44'000 km^2$). The dataset has a resolution of 1 km. It is based on a simple baseline simulation (SWE model driven by gridded temperature and precipitation input only) available over the entire period as model data, and a dataset that assimilates station observations available between 1999 and 2021, used as training data. Here, we focus on the calibration and validation procedure, and evaluate the performance of the bias-correction over the validation period. The long-term SWE record between 1962 and 2021 is produced as an example application, but its in-depth analysis is outside the scope of this paper and will be subject of subsequent work.

## 3.1 Setup, data, and validation strategy

A standard temperature index model (Magnusson et al., 2014) is used to provide a baseline simulation of daily SWE grids (the model data, see Figure 1). The model is forced with daily 1 km gridded temperature and precipitation data provided by MeteoSwiss: the TabsD product for daily mean temperature, and the RhiresD product for daily precipitation (Swiss Federal Office of Meteorology and Climatology MeteoSwiss, 2021a, b). These products are available from 1961 to the present and the baseline model is run between September 1961 and August 2021, i.e. for the period 1962-2021.

The training data is obtained using the same temperature index model, but this time in conjunction with assimilation of measured snow height, first converted to SWE, from 320 monitoring stations using the Ensemble Kalman Filter (see Magnusson et al., 2014; Mott et al., 2023, for more details). Many of these monitoring stations have only begun to measure in the late nineties, which is why the training dataset has been constrained to cover the period between September 1998 and August 2021, i.e. 1999-2021. SnowQM is calibrated and validated over the 1999-2021 overlap period, and then applied to the entire 1962-2021 period of the baseline simulation. The final corrected dataset thus provides a long-term, high resolution spatial SWE record for Switzerland based on available temperature and precipitation grids but additionally mimicking the added value due to assimilation of snow height observations.

Temperature index models often produce snow towers at certain high elevation pixels (i.e. pixels where snow does not melt during the summer and accumulates from year to year). Furthermore, monitoring stations to provide data for assimilation are not available above 2'800 m.a.s.l. and the steep Alpine topography at high elevation is poorly captured at 1 km resolution. Therefore, all pixels producing snow towers (defined as pixels having, at least once in any for the dataset, 0.2 m SWE or more remaining on the 31$^{st}$ of August) and all pixels above 3'000 m.a.s.l. are masked out in the analysis (see Figure 4). The domain where SWE grids is available is slightly larger than Switzerland, allowing to have a buffer for pixel grouping, but all analyses of the next sections are restricted to the results obtained over Switzerland only. The DEM used is provided by the Swiss Federal Office of Topography (Swiss Federal Office of Topography, 2017) and the associated slope, aspect angle, and curvature were calculated within SnowQM.

The QM is first calibrated on the odd years of the period 1999-2021 and validated on the even years of the same period (i.e pseudo-random splitting). SnowQM is run for all parameter combinations (simple hypercube sampling) given in the table 2, as well as for all $w_t$ parameters with $\Sigma P = 1$, i.e. temporal grouping only, for a total of 2190 runs. For a first analysis, the evaluation toolkit is applied to the whole country. Among the 14 metric values described in section 2.3, the summer (JJA) seasonal ME and MAE are discarded as it is not a season of interest for the dataset produced. Each parameter combination is ranked for each of the 12 used metrics separately, and a global ranking is obtained by averaging the 12 ranks.

## 3.2 Validation and sensitivity to parameters

### 3.2.1 Entire Switzerland

The performance of the QM over the validation period for the 12 metrics used is presented in Figure 5 and compared to the error of the model dataset before the correction. The ranks obtained for each parameter value are also presented in the bottom

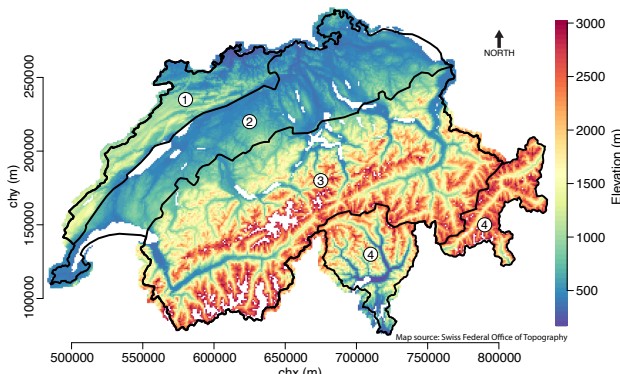

**Figure 4.** Map of Switzerland showing the topography of the country. The coordinates are in the coordinate system CH1903 / LV03 (EPSG:21781). Elevation is shown for the area where the QM is used. White areas are excluded areas (lakes, regions above 3000 m.a.s.l., regions where snow towers are created). The four regions described and used in the section 3.2.2 are represented: 1) Jura, 2) Lowlands, 3) Alps, and 4) South. Digital Elevation Model from the Swiss Federal Office of Topography (Swiss Federal Office of Topography, 2017).

**Table 2.** Parameters used for the calibration and validation runs.

| Parameter | Values |
|-----------|--------|
| $w_t$ | 10, 20, 30 |
| $\Sigma P$ | $1^{\dagger}$, 3, 5, 10 |
| $\Delta d$ | 0.1, 1, 10 |
| $\Delta h$ | 1, 10, 100 |
| $\Delta s$ | 0.05, 0.5, 5 |
| $\Delta \gamma$ | 0.1, 1, 10 |
| $\Delta c$ | 0.0001, 0.001, 0.01 |

† For runs with $\Sigma P = 1$, only $w_t$ is used,
since the other parameters have no effect.

part of Figure 5 along with the corresponding parameters' values. The best average ranking is obtained with the parameters $w_t = 30$ and $\Sigma P = 1$. In table 3, values of the metrics for the model data and for the best calibration run are shown.

The most important parameter influencing the ranking is the number of pixels used for spatial grouping, with the best results obtained without any grouping. Using a small temporal group slightly reduces the performance. When spatial grouping is enabled, the spatial distance constraint is the most important, the elevation constraint has almost no effect. Strong constraints on slope, aspect and curvature tend to slightly reduce performance. For the latter three, however, the effect is less than that of the spatial distance constraint.

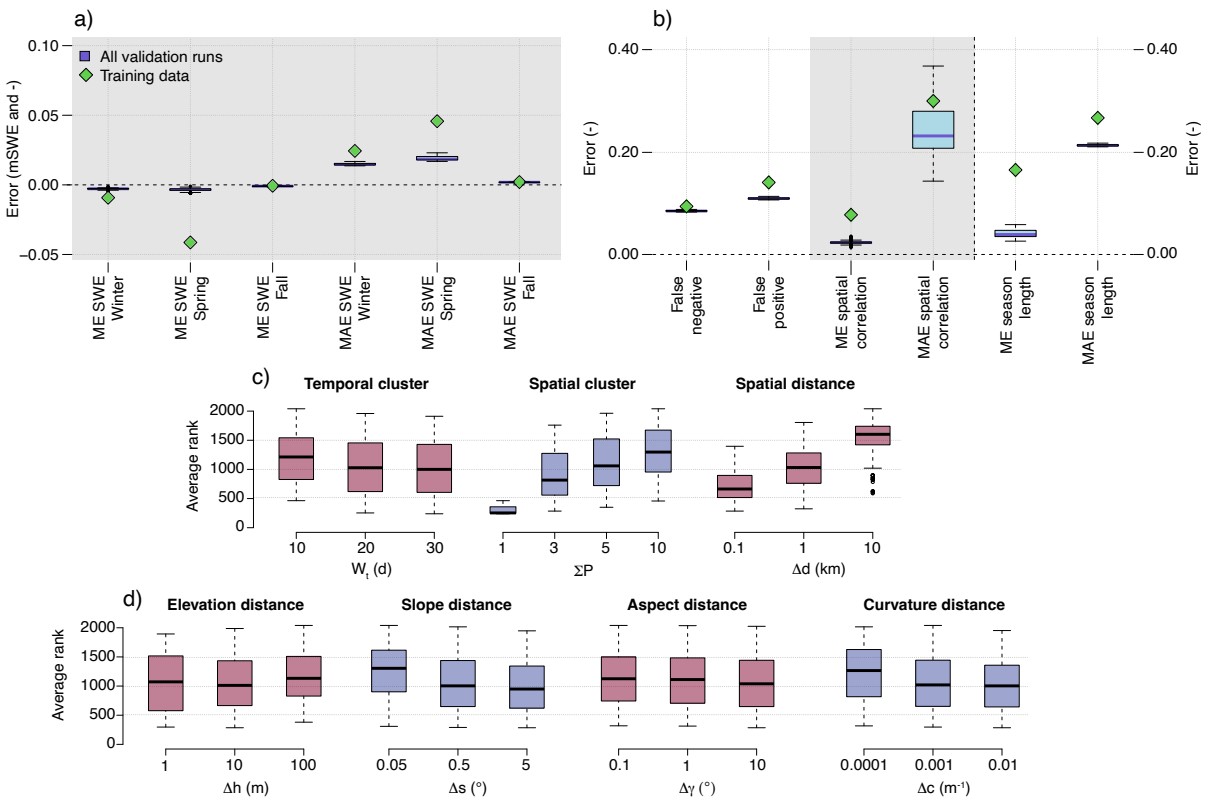

**Figure 5.** Performance of SnowQM during the validation period evaluated over the whole of Switzerland. Calibrated over the odd years and validated over the even years of the period 1999-2021. Top: Value of the metric for each calibration cycle (blue boxplots) and for the model dataset before correction (green squares). The grey areas separate the four metrics families (Section 2.3). The units of the left y-axis are m SWE for the first 6 metrics, and are relative fraction $\in [-1, 1]$ for next four metrics. Note, the last two metrics are also in relative fraction, but have different scale (right y-axis). Thick lines are the medians, boxes represent the first and third quartiles of the data, whiskers extend to points up to 1.5 times the box range (i.e. up to 1.5 time the first to third quartiles distance) and extra outliers are represented as circles. Bottom: Ranking of the corrected data for each calibration parameter.

The top part of Figure 5 shows that SWE ME after correction is close to 0 for all seasons, i.e. the bias is eliminated, independent of the parameters chosen. SnowQM is also able to reduce the error variability, as shown by the MAE of the seasonal SWE being roughly halved compared to the MAE before correction. Both spatial correlation and snow season duration errors are considerably reduced on average, but the error variability shown by the MAE on these metrics remains large despite being reduced, meaning that for some pixels and timesteps the bias is still present.

For most of the metrics, the differences between the values of the individual calibration runs are small, which means that when applied to entire Switzerland, SnowQM is not sensitive to the choice of parameters. However, the small difference in metrics can still be of importance. Figure 6 shows maps of winter mean absolute SWE value and mean SWE error before and after correction compared to the training dataset, for the best and the worst calibration runs. While both corrected datasets show

**Table 3.** Metrics values over the validation period for the model dataset before QM and for the best calibration run based on ranking (parameters: $w_t = 30$, $\Sigma P = 1$). Calibrated over the odd years and validated over the even years of the period 1999-2021.

| Metric | Units | Model dataset | Best correction | Error reduction (%) |
|---|---|---|---|---|
| ME SWE Winter | m SWE | -0.009 | -0.002 | 78 |
| ME SWE Spring | m SWE | -0.041 | -0.002 | 95 |
| ME SWE Fall | m SWE | -0.001 | -0.001 | 0 |
| MAE SWE Winter | m SWE | 0.024 | 0.014 | 42 |
| MAE SWE Spring | m SWE | 0.046 | 0.017 | 63 |
| MAE SWE Fall | m SWE | 0.002 | 0.002 | 0 |
| False negative | % | 2.3 | 2.1 | 9 |
| False positive | % | 3.5 | 2.7 | 23 |
| ME spatial correlation | % | 2.0 | 0.4 | 80 |
| MAE spatial correlation | % | 7.5 | 3.6 | 52 |
| ME season length | % | 16.5 | 4.0 | 76 |
| MAE season length | % | 26.7 | 21.2 | 21 |

a clear performance improvement compared to the one before correction, a non-negligible error is still present on average in the worst calibration run. The bottom line of Figure 6 shows the bias on the day of the validation period with the maximum SWE (4th of March 2018). It is clearly visible that at short timescale the bias is still present even in the best calibration dataset.

### 3.2.2 Regions

The sensitivity of the correction quality to the free parameters is further analysed looking at four distinct regions of Switzerland shown in Figure 4: 1) The Jura; 2) the Lowlands; 3) the Alps; and 4) the South, which are distinguished by different snow climatological regimes. The same 2190 runs as in the previous section are used, but the evaluation toolbox is applied to the 4 regions separately.

When evaluating the quality of the correction over the four regions, the good performance in terms of removing the bias in the seasonal mean SWE still applies (top part of Figure 7). For example, in the South (Figure 7d), despite a large winter and spring SWE ME in the model dataset, SnowQM is able to achieve good results by reducing these errors by about 50 and 80 %, respectively. For the spring SWE bias, a noticeable difference in performance between the chosen parameter sets is found. In the Lowlands (Figure 7b), the bias is strongly reduced for the duration of the snow season. However, in the Lowlands, the amount of SWE is always low, so this metric is very sensitive to the chosen value of the SWE threshold used to define the snow days. The bottom part of Figure 7 shows the calibration runs' ranking for each calibration parameter value for the Lowlands.

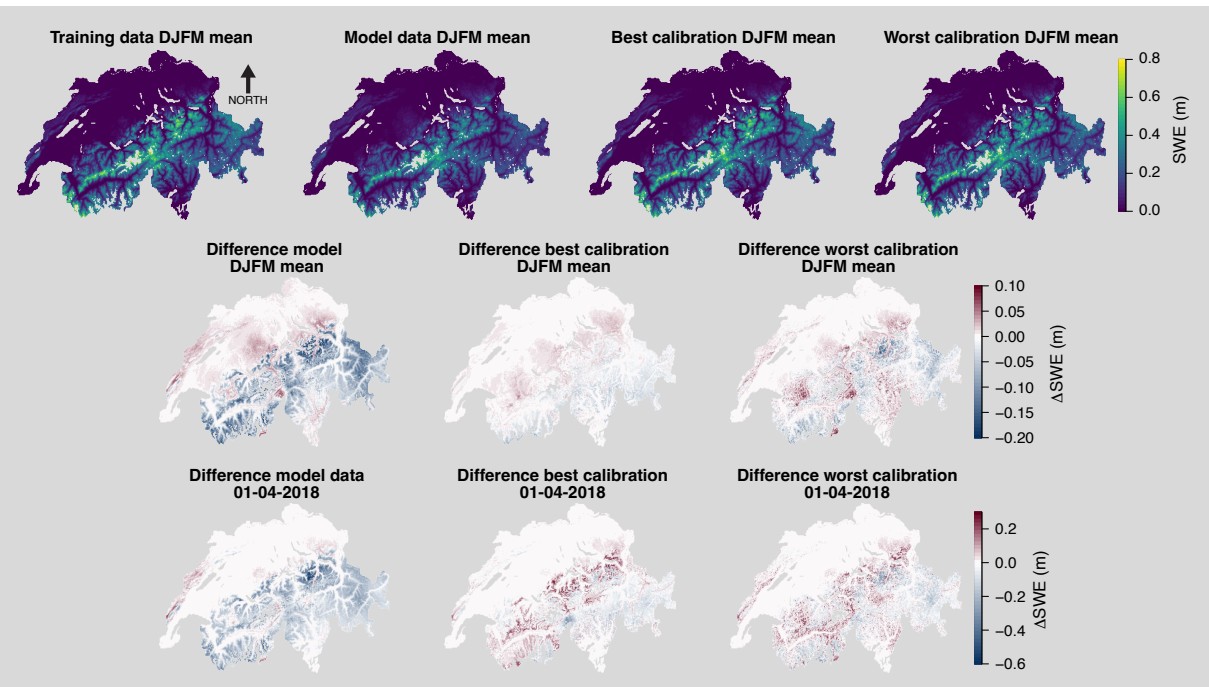

**Figure 6.** Top: Average SWE between December and March during the validation period (even years of the period 1999-2021) for the training data (using data assimilation), the model data (without data assimilation), the best calibration run (parameters: $w_t = 30$, $\Sigma P = 1$), and the worst calibration run (parameters: $w_t = 10$, $\Delta d = 10$, $\Delta h = 100$, $\Delta s = 0.05$, $\Delta \gamma = 0.1$, $\Delta h = 0.001$, $\Sigma P = 10$); Middle: Difference in SWE between December and March during the validation period for the model data, the best calibration run, and the worst calibration run compared to the training data; Bottom: Same as middle, but for the 1$^{st}$ of April 2018 (day with maximum SWE of the validation period).

Contrary to what is obtained over the whole country, using spatial grouping gives on average slightly better results in the flatter lowlands.

### 3.3 Robustness

In a second step, the robustness of the QM correction is evaluated by calibrating on years with high snowfall, and validating on years with low snowfall, and vice versa. High and low SWE winters are determined based on the SWE average over the whole country in the training dataset. The high SWE winters are 1999, 2000, 2001, 2003, 2004, 2009, 2012, 2013, 2014, 2018, 2019 and 2021. The calibration with high SWE and the validation with low SWE are an approximation of what the performance of the QM correction could be if applied to climate change scenarios, i.e. a dataset where SWE is expected to be lower than during the training period. To reduce the number of runs (to 435), and based on the observation of parameters having only a weak impact from Section 3.2.1, the following parameter values are used: $\Delta h \in \{10, 100\}$, $\Delta s \in \{0.05, 0.5\}$, $\Delta \gamma \in \{1, 10\}$, and $\Delta c \in \{0.0001, 0.001\}$ (for the other parameters, see Table 2).

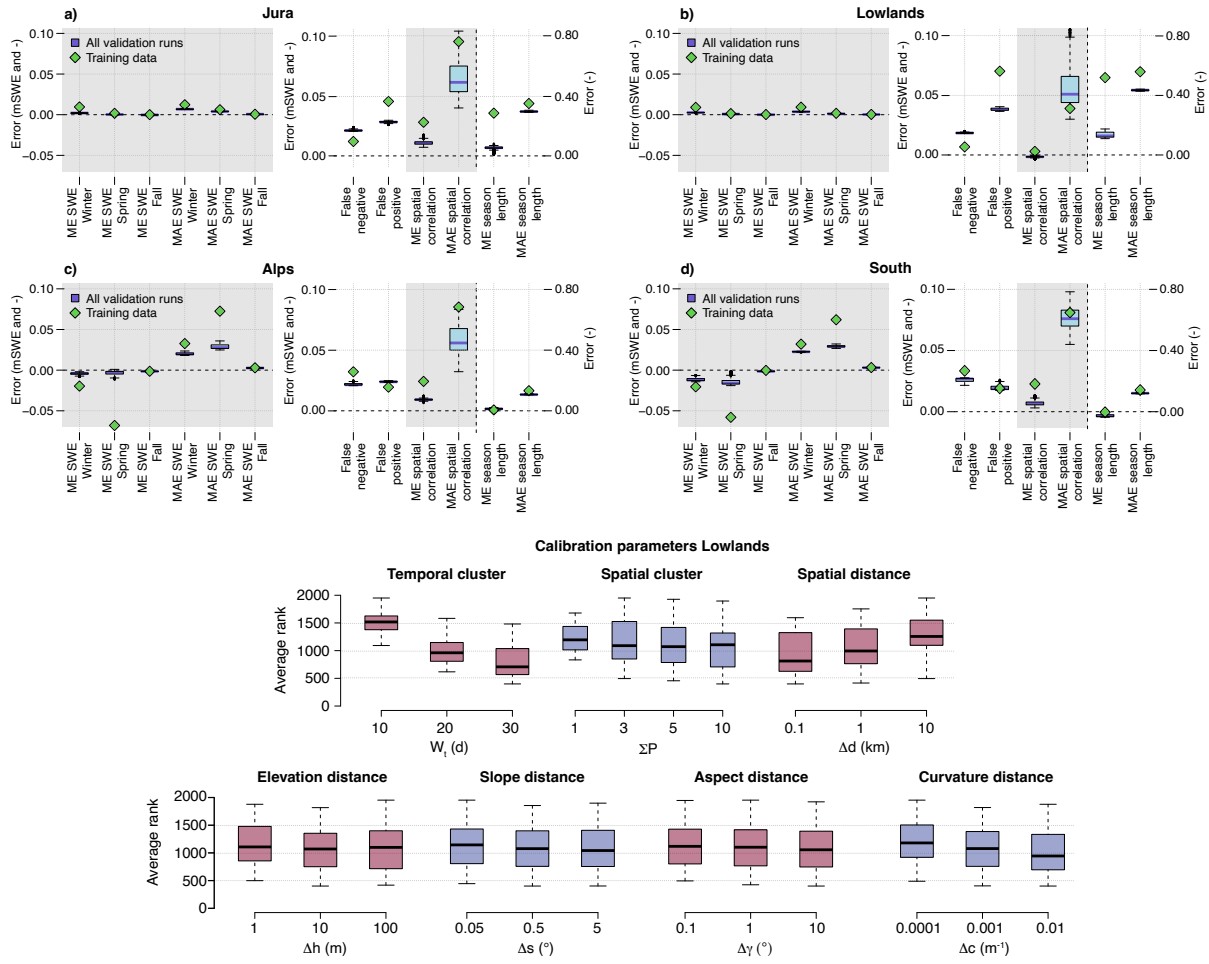

**Figure 7.** Top: Same as Figure 5 top part but for: a) The Jura region, b) the Lowlands region, c) the Alpine region, and d) the South region. Bottom: Ranking of the corrected dataset for each calibration parameter for the Lowlands region.

The correction is again very good in terms of reducing the SWE bias and, in general, there is no major difference from the performance shown in Figure 5 and in Table 3 for calibration and validation over random years. All tested sets of parameters lead to a considerable reduction of the ME in all seasons. Figure 8 shows that for the calibration over years with high SWE, best performances are obtained with temporal grouping only, as for the pseudo-random calibration. For calibration over years with low SWE, some sets of parameters using spatial grouping with a strong constraint on the distance of the grouped pixels can outperform the calibration with temporal grouping only. However, the three calibration runs with temporal grouping only still obtain good ranking (positions 20, 31 and 74 out of 435 runs).

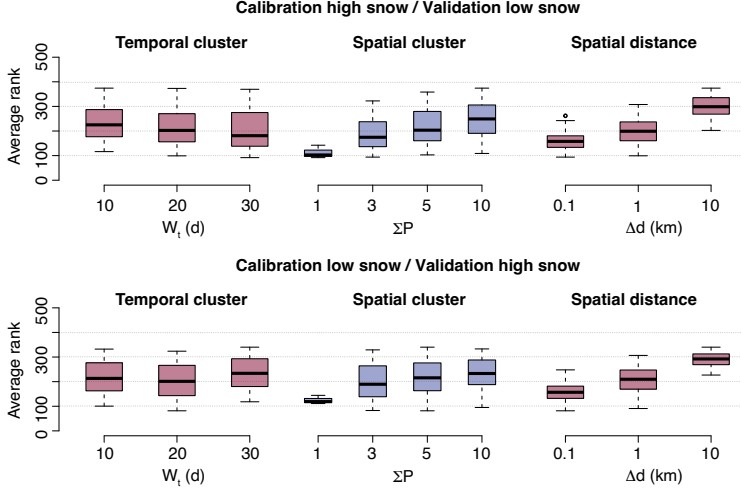

**Figure 8.** Ranking of the corrected dataset for each calibration parameter for calibration with years with high snow and validation over years with low snow (top) and for the opposite (bottom). Only parameters with a relevant impact on the rank are shown.

## 3.4 Results

The parameters giving the best performances over the whole of Switzerland ($w_t = 30$ and $\Sigma P = 1$) are used to produce the final climatological grids used this section for a more detailed analysis.

### 3.4.1 Snow water equivalent

Figure 9 shows in more detail how the SWE and SWE MAE in winter are distributed according to elevation, slope, aspect angle, and curvature before and after the correction (the ME is not shown because it is always almost zero after the correction). At all elevations, SnowQM is able to reduce the bias. Above 1500 m.a.s.l., the slight increase of the MAE (Figure 9b) still implies an important reduction of the relative SWE error due to the increase of absolute SWE amount with elevation (Figure 355 9a, note: it is not meaningful to produce the same graphs with the relative error because, as for the duration of the snow season, at low elevations only a few days of snow can produce very large relative errors). Regarding slope (Figure 9c-d), the error increases with slopes up to $10^{\circ}$, but steep slopes are mainly found at high elevation, where the SWE value is also larger. For the aspect angle $\gamma$ (Figure 9e-f), no specific error pattern is visible. Regarding curvature (Figure 9g-h), there is a considerably error in the model dataset for positive curvature (i.e. the error is smaller in the valleys). This error is importantly reduced by 360 SnowQM and the error after correction is more similar between positive and negative curvatures. Despite the larger error for positive curvature before correction (due to the larger absolute SWE value for positive curvature, see 9g, using curvature as a constraint for spatial grouping does not improve the results (see Figures 5 and 7), but the opposite is found. Probably, this is because positive curvature is correlated with elevation (summits with high curvature are usually found at high elevation) and this is absorbed in the elevation grouping.

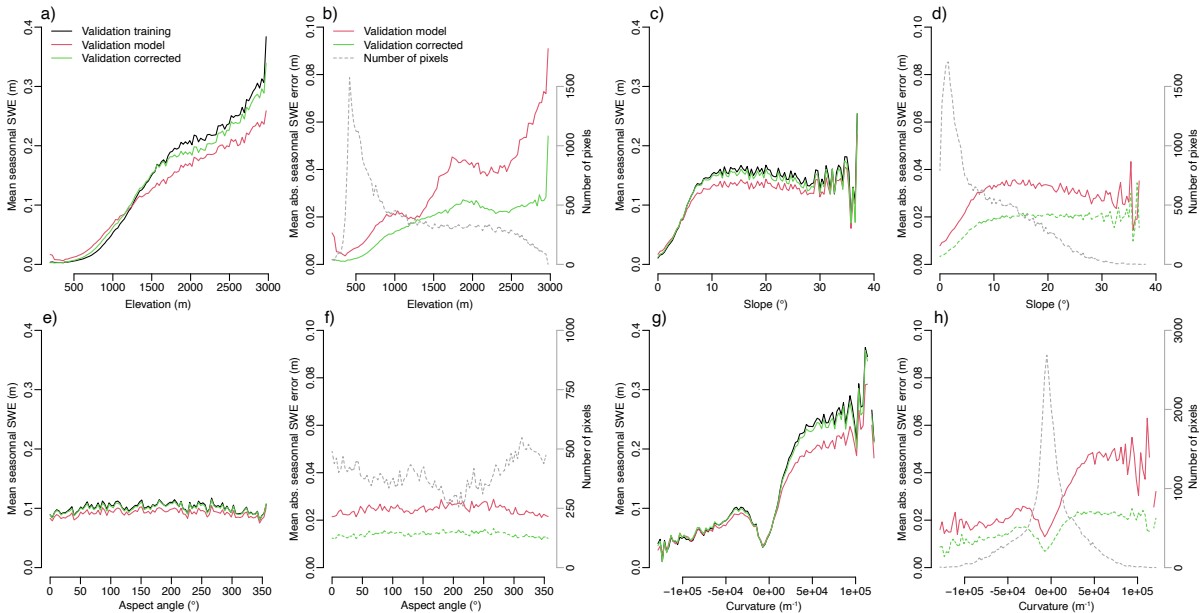

**Figure 9.** Pattern of winter mean SWE (a, c, e, g) and winter mean absolute SWE error (b, d, f, h) over the validation period for the whole of Switzerland for the training dataset (black), the model dataset (red), and the corrected dataset (green). Values sorted by elevation (a, b), slope angle (c, d), aspect angle (e, f) and curvature (g, h). Pixels are placed in 100 bins for the variable of interest (x-axis) and the mean SWE and mean absolute SWE error for each bin is shown. The grey lines show the number of pixels in each bin (right y-axis, in b, d, f, h). SnowQM is validated over the even years of the period 1999-2021. Results for the best calibration run over the whole Switzerland ($w_t = 30$ and $\Sigma P = 1$).

Figure 10 shows the training SWE, the model SWE, and the corrected SWE for 8 pixels from different regions and elevations averaged for each DOY. For each pixel, the average SWE over the period is well corrected (as also shown globally in Figures 5 and 6). For pixels with a clear bias such as Säntis or Piz Daint, where the SWE is always underestimated in the model data, the effect of the correction is clearly beneficial. However, looking into more details at pixels where there is not a constant bias but an under- or overestimation depending on the year, like in Spina (yearly data not shown in the figure), the effect

of the correction is almost negligible. Despite a small ME, the MAE for such pixels will remain large after the correction. When the SWE is averaged over regions (not shown), the MAE remains. This is thus not a local error at the pixel scale that can be smoothed by averaging the pixels together. On the contrary, it is the correction over the whole region that under- or over-estimate the seasonal snowpack depending on the season, leading to the MAE shown in Figures 5 and 9.

        Assessing the QM quality in detail at low elevation is difficult. Indeed, the magnitude for the error in SWE is far lower than

at higher elevations due to the generally low snow amounts. Consequently, SWE ME and MAE metrics do not correctly reflect these regions. Using a relative error is also not meaningful (see above). Indeed, for regions with average SWE and snow season length close to zero (usually having one snow event lasting few days or week every few years), a single snow event can produce relative errors above 1000 %. Figure 7 shows that regions with low SWE (Jura and Lowlands) exhibit a larger MAE in snow

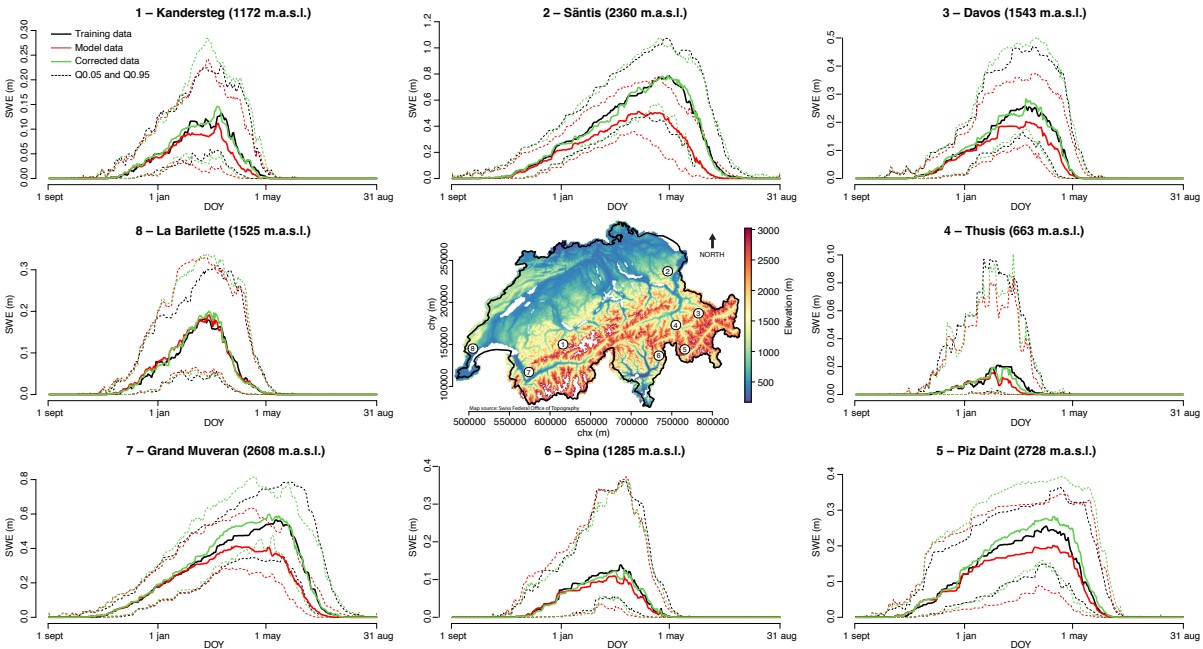

**Figure 10.** Daily SWE values for 8 locations shown in the center map. SWE from training dataset (black), model dataset (red), and corrected dataset (green). Triangles on the left part of each plots show the average over the whole time period. Results are for both calibration and validation periods (SnowQM is calibrated over the odd years). Results are for the best calibration run over the whole Switzerland ($w_t = 30$ and $\Sigma P = 1$). Digital Elevation Model from the Swiss Federal Office of Topography (Swiss Federal Office of Topography, 2017).

season length. This is mainly due to a threshold effect. Indeed, snow days are here determined using a SWE threshold of 5 mm

SWE. Looking in detail at regions with a large MAE in snow season length reveals that errors often occur when one dataset (training data or corrected data) is just above the threshold, while the other one is just below.

### 3.4.2 Climatology

To obtain the 1962–2021 climatology, the QM is trained again over the whole 1999–2021 period using the best parameters obtained during the calibration and validation phase, i.e. $w_t = 30$ and $\Sigma P = 1$. The correction is then applied over the entire

climatological period. Figure 11 shows the temporal evolution of decadal mean SWE - averaged over the months December to March - for four elevation bands for the training, model, and corrected data. Elevations below 2500 m are subject to a pronounced negative SWE trend which is caused by the well-documented step change between the 1980s and the 1990s (Marty, 2008). The step change is present in both the model and the corrected data, i.e. QM does not modify this important feature of temporal SWE variability. However, at least for elevations below 1200 m (upper left panel) the absolute change is

smaller in the corrected data, i.e. QM has a slight influence on the overall trend magnitude. At elevations above 2500 m (lower right panel) decadal variability dominates any long-term trend in both datasets. The comparison against the training data for

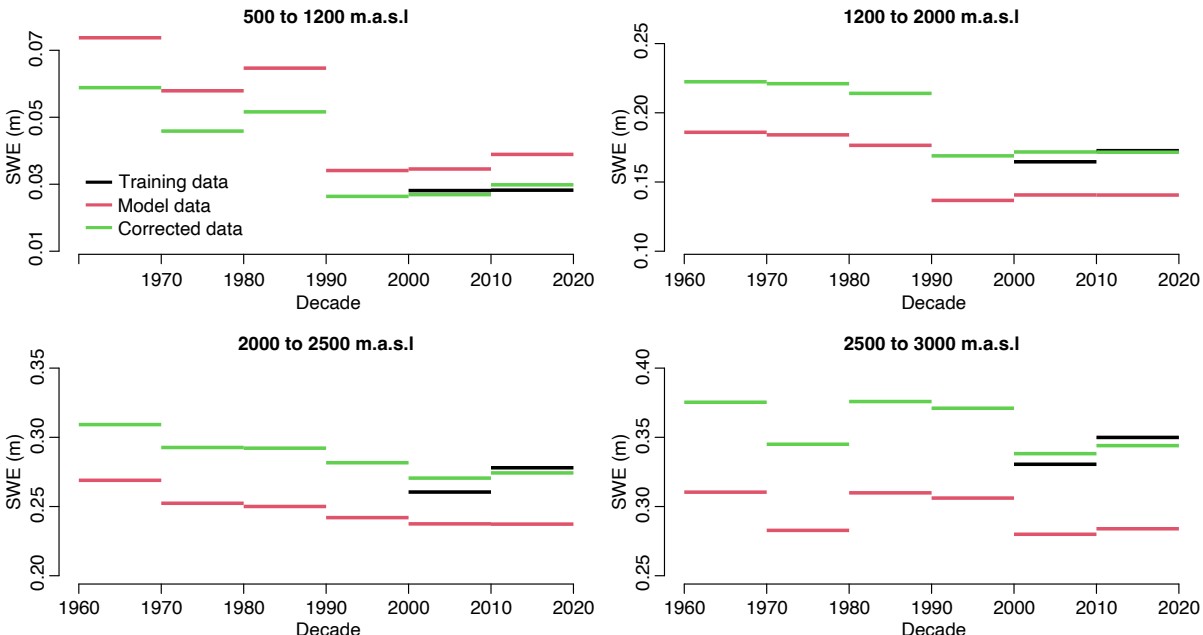

**Figure 11.** Decadal SWE average for different elevation bands. Comparison between training data, model data, and corrected model data. Results for the best calibration run over the whole Switzerland ($w_t = 30$ and $\Sigma P = 1$).

the two last decades highlights the ability of the QM to efficiently correct for erroneous SWE magnitudes in the model data at all elevations.

### 3.4.3 Computational performances and profiling

The performance of the *C++* and *R* implementations of the core functions is shown in the table 4. The performance is evaluated using the following parameters: $w_t = 10$, $\Delta d = 10$, $\Sigma P = 10$, $\Delta h = 100$, and $\Delta\gamma = 0.1$ (slope and curvature grouping are not used because they were implemented later in *C++* only). Note that the *R* implementation of some functions has been removed from the final version of the package, but is available in the GIT history (back to commit 6a228598). SnowQM runs on an 8-core Intel i7-11700KF@3.60GHz CPU with 32 GB of RAM, using *R* version 4.2.1 on *Ubuntu* 20.04.5. The QM is trained over the odd years of the period 1999-2021 and applied over the whole period. The assessment toolbox is used over both the calibration and validation periods.

The performance improvement between the single-threaded *R* and *C++* implementations is a factor of 1.7 for the quantile computation (training phase), a factor of 11 for the quantile application (correction phase), and a factor of 26 for the evaluation toolbox. As explained in Section 2.4, the improvement comes from the optimised data organisation in *C++*, which we adapt in each function to the particular data subset required. The gain is less important for the training phase, since the `quantile` function itself achieves similar performance in *R*, which uses the *C* `sort` function, than in *C++*. Using parallelisation, the

*C++* version of the code achieves almost linear scaling if hyper-threading is not used. Linear scaling was also found up to 32 threads when tested on the 64-core compute nodes of the Swiss National Supercomputing Centre (CSCS). Between 32 and 64 cores, performance increases by a factor of 1.5. SnowQM was compared with the QM implementation of the *R* package qmCH2018 (Kotlarski, 2019). In single-threaded mode (qmCH2018 is not parallelized), SnowQM is shown to be 2.5 times faster than qmCH2018. When using 8 cores, SnowQM is about 20 times faster. For the example presented here, i.e. the creation of a 60 year snow climatology over Switzerland at 1 km and 1 day resolution, any reasonably recent laptop with enough RAM to fit the entire dataset will be able to train and run the QM in about 1 hour.

**Table 4.** SnowQM performances for the *R* and *C++* implementation with different number of CPU used. See text for details about hardware and setups used.

| | *R* 1 core | *R* 8 cores | *R* 16 cores[†] | *C++* 1 core | *C++* 8 cores | *C++* 16 cores[†] |
|---|---|---|---|---|---|---|
| Training time (minute) | 270 | 48 | 38 | 158 | 20 | 15 |
| Correction time (minute) | 57 | 27 | ‡ | 5.0 | 1.0 | 0.7 |
| Assessment time (minute) | 90 | ‡ | ‡ | 3.5 | 2.7 | 2.5 |

† Hyper-threading is used.

‡ Run did not complete because of RAM limitation.

Profiling the final *C++* version of the code shows that, regardless of the number of cores used, about 80% of the time for the training phase is spent in the *C++* `std::sort()` function, and about 7% is spent writing the quantile files. Sorting the data is an essential step in quantile computation, and the existing *C++* function is already highly optimised. The function `std::sort()` is $\mathcal{O}(n \log n)$, the number of pixels used in the grouping has a direct impact on the performance during the training phase. When the correction is applied, most of the time (>80%) is spent reading and interpreting the quantile files. Increasing the number of cores helps here, despite the fact that the reading part on disk is sequential, because the file verification and data interpretation are done in parallel. However, these numbers are highly dependent on the disk access speed. In the operational implementation of SnowQM at MeteoSwiss, where NAS storage is used, the correction time is completely dominated by disk access, with no benefit from parallelism (the correction time jumps to about 15 minutes there). The evaluation toolbox does not benefit much from parallelism, as only small parts of the code have been parallelized. In fact, we estimated that the large improvement from the C++ implementation was sufficient.

## 4 Discussion

### 4.1 Performances and limitations

The analysis of the calibration procedure shows that SnowQM is robust in the sense that the overall results are only slightly affected by 1) the choice of the parameters, and 2) the choice of the calibration period, even when validating over periods with conditions different from those of the calibration. For some metrics and regions, however, the results do seem to be sensitive

to the parameters. Users should then choose the relevant metric for their application and calibrate the QM correctly to obtain the best results.

The application over the whole of Switzerland shows that SnowQM is able to efficiently reduce the more pronounced SWE bias in the model dataset at high elevations and in the valleys, which leads to a more spatially homogeneous bias after correction. The main objective of a QM-based library, i.e. the elimination of bias, is reached by SnowQM. In all configurations studied,
the average bias of SWE is always close to 0, despite the existence of considerable biases in the model data set. However, the mean absolute error can remain large even after correction. Indeed, QM is not expected to do more than a bias correction, and biases at short time scale, like on a single day or month, are not necessarily corrected (see Figures 6 and 10). Such biases can also concern entire winters at low elevated regions (see next paragraph). QM is also known to not necessarily correctly preserve extreme events (Cannon et al., 2015). Indeed, if quantile mapping is applied to a period other than the training period,
extremes not present in the training period may be encountered (in the case studied here, high SWE extreme events). A choice is then made as to how to extrapolate the quantile distribution to these new values. Here we have chosen to apply the same correction as for the 99th percentile. As a consequence, data produced with the library should not be used for extreme event analysis. Such an analysis would require an approach tailored to extreme events, as presented e.g. in Jeon et al. (2016).

At low elevations, i.e. regions with rare snow events, the use of threshold-dependent metrics as well as the analysis of relative
errors are difficult. Furthermore, since the SWE is close to zero, the total bias is also always low in these cases. This makes the correction quality difficult to evaluate with global spatio-temporal metrics. The performance of the QM in such regions shows that SnowQM is often not able to correct bias for short snow seasons. This is due to the inability of a QM model to create snow when and where the model dataset lacks snow, and is illustrated by the fact that SnowQM does not reduce the number of false negatives (see Figures 5 and 8).

The snow climatology produced shows a good agreement on decadal average with the training data (see Figure 11) and the approach is thus promising. However, the data is produced here for illustrative purposes only, and a detailed validation of the dataset using in-situ measurements and remote sensing products is required before any further interpretation of these results. Especially as quantile mapping is know to be a non trend preserving method (Maraun, 2013; Maurer and Pierce, 2014). In addition, from the points discussed above, we can already conclude that the climatological dataset can reliably be used for
analysis over long time periods, such as long-term trends, but on short time scales the QM correction is not always guaranteed to improve the quality of the data. The difficulty to properly evaluate the performance of the QM at low elevation leads to a large uncertainty for the climatology produced in these regions. However, we show that the proposed method is viable to produce snow climatology at higher spatial and temporal resolution than what exists in the literature (see e.g. Luojus et al., 2021).

Analysis of SnowQM's computational performance shows a great improvement over simple *R* implementations which are often used (see e.g. Cannon, 2018; Kotlarski, 2019), without adding complexity for users. Analysis and post-processing are also greatly facilitated by the fast *C++* code allowing to work on large grids. This opens the application of QM to larger datasets, especially as SnowQM can also be used on other variables. While many environmental software packages use only "simpler" languages such as *Python* and *R* to remain user-friendly for a community unfamiliar with compiled programming

languages, we here show that high performance can be achieved without compromising ease of use. In the form of a simple *R* interface hiding the complexity of the *C++* code, all analyses presented here were produced on a state-of-the-art HPC facility, the Swiss National Supercomputing Center (CSCS). The performances of SnowQM opens new applications which would not be possible otherwise, such as the thousands of calibration runs performed here, the application to a large set of climate change scenarios, or operational application on a daily basis.

## 4.2 Quantile mapping and data grouping

SnowQM offers the possibility to apply both spatial and temporal grouping, while QM models usually use temporal grouping only. While when applied over the whole Switzerland, the usage of spatial grouping does not improve the results. However, in the Lowlands, spatial grouping improves the correction. This is due to the flat topography of the Lowlands and the similarities between the pixels. By adding pixels, the quantile distribution becomes statistically more representative of the pixel of interest.

In steeper terrain, to include more pixels, the algorithm has to choose more distant pixels with more topographic differences, which does not necessarily increase the quality of the QM correction. In summary, depending on the topography or on the difference in the data between calibration and validation/application, the QM procedure can benefit from spatial grouping. This is an interesting finding for the QM procedure in general, since most existing studies use temporal grouping only to construct quantile distributions (see e.g. Cannon, 2018; Kotlarski, 2019).

There is is almost no reduction in the number of false negatives since using QM, snow cannot be created out of nothing, as expected and explained above. One approach could be not to work directly with SWE, but with the mass difference as determined by accumulation (solid precipitation) and ablation (snow melt), which is simply obtained by calculating the SWE difference between each time step and the previous one. The QM could then be applied to these mass difference CDFs. The corrected SWE grids are finally obtained by a cumulative sum over time of the corrected mass difference grids. For such a

variable, a random approach can be used to map the zero quantiles and choose between staying at zero or moving towards a melt or an accumulation state depending on the training CDF. The disadvantage of this method is that instead of having an independent bias at each time step, the error of the QM procedure will accumulate when reconstructing the final data set. Tests were carried out and this accumulation of error problem was found to be more important than the zero quantile problem. This approach was therefore discarded.

## 5 Conclusions

This work presents a quantile mapping package in *R* for correction of gridded snow water equivalent data. The quantile distributions are computed by grouping pixels both in space and time. SnowQM achieves high computational performance due to the parallel implementation of core functions in *C++*. Compared to a pure *R* implementation, the *C++* implementation is 2.5 time faster on a single core, and about 20 times faster when using 8 cores. The same performance improvement is obtained

when comparing SnowQM to another independent *R* implementations of quantile mapping. SnowQM can easily be extended to other variables.

A case study over Switzerland is presented. It corrects data from a simple snow model with data from an improved version of the same model to obtain a snow climatology covering a 60-year period. This case study shows that the bias in the data is efficiently removed, demonstrating that quantile mapping is an appropriate method to be used to correct climatological maps of snow water equivalent. However, limitations of the quantile mapping approach come to light as seasonal biases in snow water equivalent remain when years are compared separately. In addition, quantile mapping is shown to be unsuitable for correcting for false negative days. Depending on the regional topography and the SWE pattern, the free parameters need to be adjusted to get better results. In particular, the spatial grouping, which is a novelty of this library, is shown to improve results in flat regions. The case study also shows how such an approach can be used to produce a long-term climatology of gridded snow water equivalent. In the future, a similar method could be applied to correct bias in snow water equivalent projections of climate change scenarios.

*Code and data availability.* The source code is available on a GIT at: https://code.wsl.ch/snow-hydrology/snowqm and has been archived at: https://doi.org/10.5281/zenodo.10257951 (Michel, 2023). Installation instructions are given in the README.md. The raw data and climatology data are available at: https://zenodo.org/record/7886773 (Michel et al., 2023).

 **Appendix A: Data structure**

SWE data are stored in `vectData`. These objects are built base on *R* `list` objects. The 3D SWE input grids ($x$,$y$, $time$) are stored as 2D matrices, where x and y dimensions are concatenated into a 1D vector on the $y$ dimension of the matrix, and time is the $x$ dimension. Only some pixels are kept (e.g. if the region of interest is not rectangle, or if a mask is applied), which allows to keep in memory only necessary data and is more efficient than having the full netCDF grids. In parallel, 1D vectors are used to track the position of each pixel on the grids based on the pixel index (`x.indices` and `x.indices`) and and to convert the position on the grid into real coordinates (`x.coords` and `y.coords`). In practice, `x/y.indices` vectors store for each pixel an index pointing to the corresponding position in the `x/y.coords` vector, which hold the real coordinates.

Additional 2D or 3D grids can be stored in the same way as the SWE data, i.e having all pixels along the $y$ axis, and if needed, time along the $x$ axis. This data structure allows for quick access to one time step (pulling a column) or one pixel (pulling a line). Also, the 2D structure of the SWE data facilitate the exchange of data with the *C++* core functions. Table A1 summarizes the component of the *vectData* objects.

**Table A1.** Structure of the `VectData` R object (inherits from R `list`) object. Data are assumed to span an $m \times n$ grids, with a total of $o \leq n \times m$ pixels with data, with $t \geq 1$ time-steps.

| Component name | Variable stored | Format | Mandatory |
|---|---|---|---|
| x.coords | x coordinates of the grid cells | `array` of size $m$ | yes |
| y.coords | y coordinates of the grid cells | `array` of size $n$ | yes |
| x.indices | indices referring to the x.coords array for each pixels | `array` of size $o$ | internally computed |
| y.indices | indices referring to the y.coords array for each pixels | `array` of size $o$ | internally computed |
| time | raw timestamps for each time-step read from the netCDF file | `array` of size $t$ | yes |
| time.date | timestamps for each time-step converted into `Date` objects | `array` of size $t$ | internally computed |
| data | matrix of SWE data | `matrix` of size $o \times t$ | yes |
| dem | evation of the pixels | `matrix` of size $o \times 1$ | no |
| slope | slope of the pixels | `matrix` of size $o \times 1$ | no |
| aspect | aspect of the pixels | `matrix` of size $o \times 1$ | no |
| curvature | curvature of the pixels | `matrix` of size $o \times 1$ | no |

*Author contributions.* AM developed the model code, performed all the simulations, and wrote the initial paper draft. TJ provided all the data from OSHD. AM, JA, SG, SK, and CM gave input to the model and reviewed different model versions. SG, SK, and CM designed the project. All authors reviewed and commented on the manuscript.

*Competing interests.*   The authors declare that they have no competing interests

*Acknowledgements.*   The authors wants to acknowledge Jan Rajczak, Michael Lehning, Regula Mülchi, Sarina Danioth, and Hendrik Huwald for their contribution though discussion during this project. The vast majority of work was performed with open and free languages and software (mainly C++, R, and bash, and countless libraries). The authors acknowledge the open-source community for its invaluable contribution to science. The simulations were performed on the Piz Daint supercomputer of the Swiss National Supercomputing Center (CSCS, grants
s938 and s1015). The CSCS technical team is acknowledged for its help and support during this project.

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
