# Peer review of "SnowQM 1.0: A fast *R* Package for bias-correcting spatial fields of snow water equivalent using quantile mapping"

_Geoscientific Model Development, 2022_

## Author Response (AR1)

Dear Reviewers and Editor,

Dear Marianne Cowherd, dear Michael Matiu, dear Steven Phipps,

We appreciate your insightful and constructive comments on our manuscript. They have allowed us to present a much improved version of our manuscript and package. We would also like to apologise for the time it has taken us to respond. Please find below a detailed response to each of your comments and concerns, along with the changes we have made to the manuscript.

The reviewers' original comments are in black and our responses are in blue. Note that we have taken the opportunity of the manuscript revision to update some relevant literature and provide a bit more context about SnowQM. These additions are listed in our response below.

Please note that the page and line numbers in this document refer to the updated version of the manuscript without track changes.

Yours sincerely,

Adrien Michel, on behalf of the writing team.

**Reviewer 1 - Michael Matiu**

*Michel et al. supply a package to perform quantile mapping for spatiotemporal grids. It's usage is specifically designed for snow water equivalent maps, but can in principle be applied to any variable (as claimed by the authors). Compared to existing approaches, the author's implementation offers a flexible way to include temporal and spatial neighborhoods in the QM training step. Finally, it is designed to make the most of lower-level languages and parallelization to achieve high-throughput for large spatiotemporal data.*

*Altogether, the combination of temporal and spatial neighborhoods with a good flexibility and fast computational characteristics present a novel and worthy contribution to existing QM implementation. Even though, a few clarifications are needed before publication (see below).*

**Major points**

- Introduction could be improved: Currently it focuses a lot on what you did (which is the job of the other sections of the paper). And since one major strength (I assume?) of your package is the spatial component, you could also provide some background on spatial bias correction approaches.

  This was indeed missing. We have added the following paragraph (see P2L43-48):

  "When quantile mapping is applied separately to each pixel of a grid as is commonly done (see e.g. CH2018, 2018), the spatial structure of the data may be poorly matched to that of the observations. There are several approaches that try to overcome this limitation, many of them based on the shuffling and reordering of the data introduced by Clark et al. (2004). In SnowQM, we propose a parameterisation for the construction of the quantile distributions by temporally and/or spatially grouping the data for each pixel and evaluate how it helps to preserve the spatial dependence of the data, which to our knowledge is not present in the literature"

- Language: Consider changing the language throughout the manuscript. Clusters refers to distinct (non-overlapping) groups, while what you do in your approach is more a temporal (and spatial) moving window or neighborhood approach.

  You are correct; we do not perform clustering, as this is commonly understood (see e.g. Fiddes, 2014). The words "clustering of the pixels" has been replaced by "grouping of the data". We also now explicitly explain how this differs from a clustering (see P7L160-163) :

  *"Note that this approach differs from traditional clustering as in Gutierrez (2004). SnowQM does not create clusters of similar pixels to gain computational efficiency by performing a single calculation for the entire cluster. Instead, it takes data from a group of similar pixels to construct a more representative quantile distribution for the pixel of interest, but each pixel uses its own group."*

- Re-usability and applicability:

  o You could consider simplifying the installation command to a one-liner devtools::install_git("https://code.wsl.ch/snow-hydrology/snowqm")

    Thank you for the suggestion, we have updated the readme.md file accordingly (see commit 4420908a).

  o Since QM is often used not only for bias-correction but also downscaling (e.g., CH2018), what are your takes on this? Could your package also be used in this regard?

    The answer is yes, but not without some reworking of the code. As it stands, the code is designed to work with grids of similar resolution. However, since everything is stored in a custom `vectData` object format which is easy to handle, it would be easy for a user to construct this object to have correspondence between grids of different resolutions and use the SnowQM core for downscaling. This would not require any modification to the C++ core of the package. This is now mentioned in the model Vignette (see commit 9abbd6ec).

  o If you are interested that your package is re-used and applied in the community, you could consider updating the code documentation and vignette, and providing examples with some example data. Also posting the package to CRAN is great way to boost usability.

    The documentation has been updated and a vignette has been created (see final commit 00348663). Regarding CRAN, unfortunately the lack of native OpenMP support on MacOS makes it difficult to upload an OpenMP dependent package to CRAN. In fact, packages should compile on Linux, Windows and MacOS in order to pass CRAN testing. We estimate that the extra work required is not worth it. But thank you for the suggestion.

  o How easy would it be to apply snowQM to other variables like precipitation and temperature? You claim it is theoretically possible, but how much effort would this mean in practice? Furthermore, do you have the option to correct, e.g., wet-day frequency?

    In practice, no effort is required; just provide NetCDF grids with another variable (the name of the "data" variable is a parameter of the function call, everything is now detailed in the documentation). However, there are no options other than those described in the paper (e.g. wet day frequencies).

o   Since the spatial neighborhood approach does not seem to provide substantial benefits, could it be instead used to speed up computation? For example, by correcting similar pixels simultaneously? That is, in a true clustering sense, where clusters of similar pixels are trained and corrected together.

This could indeed be envisaged. However, clustering (as this is commonly understood, see e.g. Fiddes, 2014) is more often used in downscaling, where large pixels are downscaled into smaller pixels, and the smaller target pixels can be clustered together to increase efficiency (see, for example, Gutiérrez, 2004, and Fiddes, 2014) without any real loss of information. Since we have grids of similar resolution, 'real' clustering would imply the loss of some spatial information.

- As repeated throughout the manuscript, a main strength of your implementation is its speed. However, what I don't understand is why the C++ variant is 12-27 times faster in the correction phase, where 80% of the time is spent on harddisk operations (as per your profiling)? While in the main CPU process of training the benefits of C++ are more in the order of 2.

The profiling numbers presented here only refer to the final C++ version (this was referred to as the "final version"), so it cannot be used for comparison with the R version. Also, in the first version of the manuscript, we said wrote "spent reading the quantile file", which is the disk operation and the CPU time to process the data. This was indeed not clear.

Since both you and the second reviewer had many comments on this section, we have largely rewritten sections 2.4 and 3.4.3 and included the clarification you requested, see P8L234-256 and P20-21L395-424.

- Related: You could consider being more neutral when discussing computational benefits of pure R (or python) versus C++, because the impression arises that C++ is the only way to achieve high throughput. There is more to it: easy parallelization of standard R (e.g., with the foreach package), RAM and harddisk bottlenecks. And finally, there is the trade-off between between higher- and lower-level languages in how easy it is to understand and modify code. Nevertheless, it is true that re-writing slow components in C++ can significantly improve computations, which is quite standard practice for R programmers which deal with computational bottlenecks, btw.

Thank you for your comments. First, yes, this is standard practice for developers. However, in environmental science most people only know a scripting language like R and are not necessarily aware of the potential that proper profiling and optimisation can offer in terms of scalability of the models. This paper is aimed at both more experienced developers and environmental scientists interested in our SWE product. In doing so, we have chosen to emphasise some aspects that may be obvious to some readers. This is also what we explain on P10L252. We wish you would agree with this approach.

Regarding the first part of your comment, we have mostly rewritten 2.4 and added details about bottlenecks, the reason why we chose C++ and not R for parallel computing, and the trade-offs of such a choice, see P10L239-243 and P10L245-249.

**Minor points:**

- L3: "Accurate…" seems a bit overstated, or not clearly articulated. What do you mean by accurate? I guess this statement depends strongly on spatial and temporal aspects. Please consider rephrasing.

  Thanks you. In fact, "accurate" is not really meaningful and necessary there. It has been removed.

- L9: reduces bias with respect to what?

  The sentence has been rephrased as (see P1L4-6):

  "This study presents a \textit{R} package, SnowQM, designed to correct the bias in long-term spatial snow water equivalent data compared to a shorter-term and more accurate dataset, using the more accurate data to calibrate the correction."

- L11: did you mean heterogeneous instead of homogeneous?

  No, we really mean "homogeneous" here. In fact, as explained in Section 4.2, spatial grouping is beneficial in the lowlands, while it does not improve bias correction for the Alpine region.

- L39: not quite accurate. Many functions in base R are coded in C++, so if these functions are used, then R not slower than C++.

  This is true. R is written in C, so the basic R functions are quite fast (we tested and the R quantile function is as fast as our C++ implementation). We have rephrased the sentence to (see P2L51-53):

  *"Finally, SnowQM is distributed as a user-friendly R library, but the core of the model is coded in C++ to achieve significantly higher serial and parallel performance than R".* There are other advantages of C++ for manipulating large datasets, where typed C++ vectors are faster than R objects, but the main improvement and motivation for the C++ kernel is the limitation of parallel computing in R and the associated memory issues.

- L89: Maybe mention that besides using the ECDF as approximation to F, one could also use parametric variants (which Gudmundsson et al. 2012 assessed), even though ECDFs are usually preferred (cf. reviews by Maraun and others).

  This is what we mean by "based on the work of Gudmunsson et al. (2012)", i.e. we have chosen an empirical rather than a parametric approach. We have clarified the text as follows (see P5L112-115):

  *"There are a number of different approaches to approximate $F_{mo}$ and $F_{tr}^{-1}$, which can be either empirical or parametric (see, among others, Maraun et al., 2016). Based on the work of Gudmnunsson (2012), the functions are approximated using the empirical quantile distribution, i.e. the data are sorted in ascending order and simply separated into n-1 bins of equal number of points, where n is the number of quantiles used."*

- L146: I think this is a limitation if the correction is performed day-by-day. If performed at longer time slices, one could also use the same probabilistic approach used for precipitation. Which conserves frequency (snow vs snow-free period) and precip amounts (total SWE). For instance, with reshuffling the time series after the QM step?

  Yes, this could be an approach to get the correct seasonal snow cover. However, our method will be implemented operationally at MeteoSwiss on a daily basis, i.e. the dataset will be extended every day during the winter season with the reanalysis snow simulation of the previous day (note that we are aware of the many limitations of our product and the analysis will only be performed on temporally and spatially averaged data). In this respect, we do not want to make a statistical frequency adjustment that might differ from reality. In addition, if such an approach is applied pixel by pixel, it will be difficult to maintain spatial consistency.

- L154: same as above. One could consider also the aim of QM to transfer the seasonal properties (length, total accumulation), then "creation" or "removal" of snow seems less an issue.

  Indeed, but again this prevents any operationalization of the method. I.e. the data are only available at the end of the snow season and we lose the opportunity to assess the state of the snow cover in relation to the climatology during the winter. Based on your two comments, we decided to add a paragraph in the introduction mentioning the planned subsequent operationalization (see below and P2-3L55-64), which includes some specific requirements. This important information was indeed missing.

  *"The development of SnowQM is part of a joint project between the Swiss Federal Office of Meteorology and Climatology, MeteoSwiss, and the Swiss Federal Institute for Snow and Avalanche Research, SLF. The aim of the project is to obtain a long-term climatology of snow water equivalent (for research purposes) and snow height (for public purposes) that is operationally updated on a daily basis. In this project SnowQM is used to produce the snow water equivalent data and the model SWE2HS (Aschauer et al., 2023) is used to convert the snow water equivalent to snow height. The operational use of SnowQM adds some constraints to the development (e.g. working on daily data rather than already temporally aggregated data, although climatological analysis is not performed on a daily timescale). The full model chain was tested during winter 2022-2023, will be operational internally during winter 2023-2024, and the automatically generated operational analysis (plots) will be publicly available during winter 2024-2025. The full publicly available dataset will be updated at the end of each winter season."*

- L194: Not clear what your intent is with this spatial metric?

  To see if the model is correctly capturing the number of snow days. At high elevations, this is redundant with the length of the snow season. However, at lower elevations with intermittent snow seasons, the length of the snow season is not meaningful, justifying the addition of this metric.

- L213: what do you mean by homogenized?

  We meant consistent over time, but this word is not necessary and has been removed.

- Fig5: It would make more sense to split the figure into two sets of metrics: pos only, like all MAE and the FP and FN counts; and then the ME metrics which can be both pos and neg. In this way you could make better use of the y-scale.

  Thank you for the suggestion, see updated Figure 5 below. We also updated Figure 7 accordingly (see below).

[Figure]

Updated version of Figure 5

- L249: Unclear how you determined the ranks. Also ranking in general discards all information on magnitude of differences. Wouldn't it make more sense to look also at absolute or relative gains in performance?

  We tested with a relative ranking. That is, each run and metric is ranked between 0 (worst run) and 1 (best run), with the grades being a linear scaling based on relative performance compared to the best and worst runs. The exact ranking differs, but the main conclusions (temporal grouping is the most important parameter, spatial grouping improves model performance over lowlands) remain the same.

  Regardless of the ranking chosen, all metrics have the same weight, which is a major limitation. Being the best run for a metric where all runs are very close to each other has the same weight as being the best run for a metric that has an almost similar value for all runs.

  This is a highly non-trivial question and, to our knowledge, there is no satisfactory solution other than a purely subjective choice such as we have made here. At MeteoSwiss, for example, the operational

numerical weather model is verified with 7 to 13 metrics, depending on the variable, with no preference for any metric.

[Figure]

Updated version of Figure 7

- L280: What is with the other regions?

  They show similar characteristics to Figure 5, so we do not them it in the manuscript.

- L307: you could still do relative errors for all values above a threshold (mean SWE or elev), if you think it's necessary

  Yes, that could be done, thank you for the suggestion. However, in this particular case, it does not add any information to what we already see in Figure 9 and what is said in the text, so we decided not to add this figure to the text. We also decided to keep the text intact and not overload it with more detail here.

- Fig 10 is difficult to read. Please consider reducing the number of years (or increasing the width of each subplot timeseries, or both, whatever works good), so as to better the see the different lines.

  Thank you for the suggestion, see updated Figure 10 below.

[Figure]

Updated version of Figure 10

- L330: I could imagine the issues at low elevation might also be related to your choice of a restricted time period, which might not capture enough variability to derive ECDFs properly.

  Yes, this is also shown in the lower panel of Figure 7. At low altitude, a larger time window improves performance, which is not the case globally (see Figure 5, bottom panel). However, we tested with even larger time windows (45), which did not improve the results.

- L352: how was the 57 derived? Comparing 1 core R to 8 core C++? Is that fair?

  This is indeed the wrong value, it should be 11 (57 to 5 minutes), this has been corrected on the revised Section 3.4.3 (see P20L403). Thank you for catching it.

- L379f: Not sure about this statement. The strength of QM (compared to simple adjustments like mean and variance) is that it corrects the whole distribution, which includes extremes. The paper you cited concerns more trends in extremes using century long climate simulations.

  While extremes are corrected, extremes can occur outside the range of data used to train the model. Therefore, we have no guarantee of how well they are corrected. There are different approaches, i.e. keep quantile 99 as we do, or extrapolate the quantile correction based on the higher quantiles. We have rephrased for clarity and added references (see P22L438-443).

- L382: What about splitting the evaluation by elevation?

Yes, this could be done. However, this introduces a new problem of spatial inhomogeneities at the boundary between selected elevation bands, which needs careful verification. For the climatological product, we then decided to use the same parameters everywhere. However, as we say in the text, if users are only interested in some regions or elevation bands, they are encouraged to tune the parameters accordingly.

In addition, and for full transparency, it would require at this point a full rerun of all the calibration runs (with a significant computational requirement). Indeed, more than 3000 runs were performed, producing each datasets of 10 GB. We did not have the opportunity to store on the long term the more than 30 TB of data produced. All the metrics were computed at the end of the run, and the underlying dataset were directly deleted. A re-run is still possible, but we would perform it only if really needed.

**Reviewer 2 - Marianne Cowherd**

The authors present an R package for computing quantile mapping based bias correction of snow water equivalent using Switzerland as a test case. The package includes C++ implementation of routines and OpenMP parallelization for speed within the R package. The package also includes built-in evaluation functions. QM is an important bias correction method with many possible applications and this package is a positive contribution to the accessibility of QM with both spatial and temporal windows used for the clustering.

Major comments:

The phrase "snow cover" is sometimes used when you may mean snow water equivalent. For example L37: "tailored towards snow cover" when it appears to be tailored to snow water equivalent.

Thank you. This is indeed an unfortunate formulation. We did reword it in the whole manuscript when needed (note in some instances snow cover was appropriate).

There is a long discussion of speed. However, it would be good to more clearly separate the speedup achieved from parallelization vs. from the C++ implementation, especially as the user base will likely have varying access to large numbers of cores when running the code.

Thank you for the feedback. See our answers to the first reviewer about similar questions. We have improved the whole Section and now cover clearly the points you mention (see updated Section 3.4.3)

A longer discussion of the role of spatial clustering would better explain some of the results. It seems that the proposed best-ranked parameterizations have small clusters and therefore weak use of the spatial aspect. Can the clusters be overlapping with clusters from other pixels? It seems that they can, since each pixel is computed independently, which is not really a common way to use the word "cluster". If you do use the spatial term for clustering and do not repeat a computation for each pixel within the cluster and they are nonoverlapping, this could be a future way to improve speed (at the risk of reduced accuracy).

See the response to the first reviewer who made the exact same point. In summary, the word "cluster" has been removed for clarity (we now use "pixel grouping") because we are not doing clustering in the

sense that QM is not computed only once for a cluster of similar pixels. What we do is that for each pixel we also include data from similar pixels to have more points in the distribution, but each pixel has its own group.

Regarding the point you raise about our best-ranked parameterization favouring small groups, let us give a few more details. First, we show that in steep terrain, the grouping is not improving the results. As explained above, in steep terrain the grouping has to relax more the constraints to find pixels. Thus, the pixels included in the grouping are not so similar in the end (recall, the grouping is done pixel per pixel, and for each pixel separately the constraints are slowly relaxed until enough pixels are found). In flat terrains, pixels included in the group are more similar and thus more efficiently helping the quantile mapping. Since the metrics chosen are based on absolute error (opposed to relative error), the error in Alpine regions has more weight, despite covering a smaller area. Therefore, when the metrics are computed over the whole country, smaller groups are preferred. As discussed in our response to the first reviewer, there is indeed a subjective aspect in the choice of the metrics. As we mention on the manuscript, depending on the application the metrics should be carefully used.

A solution could be to not fix the group size and relax the constraints, but to fix the constraints. The consequence would be larger groups in the lowlands, where it is beneficial, and smaller groups in steeper terrain. This was the first approach implemented. However, as mentioned in the manuscript, we discarded it because this can easily lead to huge groups in flat terrain (many thousands of pixels), really slowing the execution (and not improving the model quality as this point).

Some of the discussion focuses on the areas with low or zero snow, however many of those areas are not particularly of interest for snow maps because there is no snow there. While areas in and around the snow line would be of interest, there is a risk that never-snow regions could inflate accuracy statistics because they will always perfectly match at 0. Showing elevation-stratified performance is helpful for this, but it would be good to be clear in some of the error metrics about showing only snow areas. Would it be possible to include historical snow values as an additional clustering criterion in a future version?

Thanks for the comment. Firstly, low snow regions are still of interest to some users. Indeed, in the Swiss case study, the vast majority of the population lives in the lowlands, where there is no snow for most of the winter. However, snow is still relevant there as snow covered days have a large impact on the population, e.g. for transport. Therefore, it is important to have good statistics on the development of snow in this area.

Yes, we agree that "always no snow" pixels artificially improve the results. However, as explained above, having the correct true negative in the final product is of interest. However, for applications such as hydrology, it probably makes more sense to look at the snow maximum only for snow-covered pixels. This brings us back to the subjective part of the metrics. For SnowQM, an important point is the perspective of operational application over all regions of Switzerland, explaining some of the choices made (we have updated the introduction to emphasize this, see P359-61).

Your suggestion to use past snow cover to group pixels is relevant. In fact, this is what we are trying to mimic somehow by using topographic information. We will definitely keep it in mind for a future update of the model.

A discussion of potential expansion to other quantities other than snow and to other regions would be a welcome addition.

We have added an explanation in the Vignette and rephrased the introduction and conclusion to really emphasis on this feature (see P3P71-72). See also our response to the first reviewer about this point.

Minor comments:

L8: parallel computing is possible in any language; the choice of C++ for speed is parallel to the choice to implement in parallel

Yes, we agree with your point. However, note that parallel computing in R is very unfriendly because it involves a full copy of memory to each thread. Efficient implementation therefore requires an interface to an external language. There were indeed other options. We chose C++ because the Rcpp library provides a simple interface and because the main author is familiar with it. We have rephrased the sentence to acknowledge that parallelization does not necessarily require C++ (see P1L8).

L11: if homogeneous then this is a limitation for snow water equivalent, especially in mountainous terrain (including Switzerland). Possibly heterogeneous?

No, we mean homogeneous (see also our response to the first reviewer). What we show is that our attempt at spatial grouping does not really improve QM in steep terrain (mainly because "similar" pixels in terms of elevation and orientation can be quite far apart, and because weather conditions can vary significantly over short distances across the Alps), but does help in flatter terrain. As our application is focused on the Swiss Alps, we do not use spatial grouping in the final product. However, we believe that this experiment is worth mentioning (even negative results are relevant) and is useful knowledge for QM in general. Indeed, this approach could be of interest for SWE QM in a smoother terrain such as eastern Canada.

L26-48: the detailed description of the attributes and benefits of SnowQM are better suited for a discussion or conclusion. Especially L35-14 would be better at the end of the paper. Consider replacing with more detailed literature review, for example a longer introduction to previous QM snow efforts (e.g., Jorg-Hess et al. (2014) as cited in this section already) would be important to discuss. Consider Matiu and Hanzer (2022).

We think that the reference given in the introduction have their place tthere. However we agree that a more detailed discussion on previous similar studies (namely Jorg-Hess et al. , 2014, and Matiu and Hanzer, 2022, which are the only similar studies as much as we know) was missing and we added a paragraph in the introduction see P2L36-43. In addition, we added at several locations in the manuscript more discussion and reference to the literature (see P22L439-443, P22L443, and P24L479-480).

L50: acronym SWE can be introduced at first use in the main body of the text

SWE acronym in introduced at P3L73. Note we do not use acronym in the introduction.

L50-55: suggest re-phrasing this paragraph to decrease the number of parentheticals. As written, it is somewhat difficult to follow

See new paragraph (see P3L74-76).

L65: does the package require daily data? It should require the inputs and outputs to be at the same temporal resolution

In theory, yes. However, since the QM is calculated for each day of the year, the library assumes daily data and calculates the day of the year based on the date to make the temporal grouping. This is fine for any time step as long as a date is provided. However, this can lead to undesirable behaviour. For example, if hourly temperature grids are provided, each time step will be treated separately, but the temporal grouping will combine the time steps of neighbouring days, whereas the user will probably prefer to group different days for the same hour of the day.

In other words, there is no technical limitation, but the correct handling of sub-daily information would require a more complex grouping, which is not yet implemented. In addition, the time window given as a parameter is the number of neighbouring time steps to be used (this is how it is implemented), which could be confusing. Therefore, we prefer to keep the information about daily data in the manuscript, but have added a note about it in the package Vignette (see commit 8042df13).

L66: the reference of "they" is ambiguous

See correction (see P4L89).

L75: lowercase n in netCDF

Thank you, we corrected all occurrences.

L94: more explanation of this

We have added some details, see new text (P5L119-121):

*"Indeed, once the sample has been partitioned into n-1 bins, the question of where exactly to take the quantile value between the max value of a given bin and the min value of the next bin is subjective. Our implementation corresponds to the 7th definition in Hyndman and Fab (1996), from Gumbel (1939), which is a linear interpolation between the two closest values using the quantile position."*

L109-11: split into two sentences to avoid long parenthetical: "Temporal clustering is straightforward: given a time window parameter wt, the CDF for DOY is calculated using all SWE values for the pixel of interest in the time interval i±wt of each year. However, spatial clustering is more difficult."

Thank you for the suggestion, we implemented your proposition (see P6L135-137).

L199: cite the R core team at first mention of the R programming language, in the introduction.

Indeed, the manuscript has been modified accordingly

L209-210: "laptops, workstations, and HPC implementations" is vague – the goal here is to demonstrate testing on multiple operating systems (previous phrase) and architectures, and specify (briefly) architecture information. Name the HPC system and state the number of cores/nodes used for the laptop, workstation, and HPC. "Installation and usage were easy" is subjective, remove.

*In the reworked Section 2.4 (the first reviewer also provided comments on this section, leading to a full rework), we implemented the corrections you suggest (see P10L254-256).*

L213-216: sentence is long, split into 2-3 sentences for clarity. "for entire Switzerland" – "for the entire area of Switzerland"

*The sentence has been split in three sentences (see P10L258-261), thank you for the suggestion.*

L226-235: how are the snow height measurements used to improve the SWE time series? Is it a reanalysis or direct injection to a model?

*Snow height measurements are first converted to SWE and assimilated daily using an ensemble Kalmann filter. The model has been run in the past for reanalysis and is operationally updated daily at SLF. Note we slightly updated the paragraph for clarity and added a recent reference about the model (see P11-L271-273).*

L235: do you have a strict definition for how to identify snow towers in the model?

*Yes, the definition used is: pixels having, at least once in any for the dataset (model and training), 0.2 m SWE remaining on the 31$^{st}$ of August. Note the SWE amount can be specified as a parameter when the mask is computed (see model Vignette for an example). We updated the manuscript accordingly (see P11-L283-284).*

L255: "really small" is vague

*We agree. We realized that the whole paragraph was not clear. We propose the following new version (see P12L301-304):*

*"[…] When spatial grouping is enabled, the spatial distance constraint is the most important, the elevation constraint has almost no effect. Strong constraints on slope, aspect and curvature tend to slightly reduce performance. For the latter three, however, the effect is less than that of the spatial distance constraint."*

Figure 5: Add letter labels to make referencing more clear. For the spatial cluster parameter, is there a physical meaning to a value of 0? The listed values start at 1 in the table

*The value 0 for spatial cluster (now called spatial group size) should be 1, i.e. no spatial grouping (only 1 pixel, the pixel of interest, in the spatial group). This has been corrected in the figure (as well as in Figures 6 and 7), thank you for pointing this out. Note (and we have added this explanation in the Vignette) that the value 0 can be used in SnowQM to use constant distances when creating groupings (i.e. the group is not created with a target group size, but all pixels that meet the distance requirement given as a parameter are included in the group, resulting in groups of different sizes depending on the location).*

L263: if "significant" does not mean statistically significant, find another word to describe the impact of the difference

"Significant" has been replaced by "of importance".

L280: quantify this difference. This seems to contradict the previous finding that the best parameterization had a cluster size of 1 pixel. If that is not the case, clarify the relationship between the sigma P parameter and the spatial factor here.

In general, i.e. over the whole country, best results are obtained with a spatial grouping of size 1. However, over the lowlands, spatial grouping does help to improve the results. This is discussed later in the Discussion section (see P23L472-476). Indeed, sigma P is the size of the spatial group, and part of the confusion probably comes from the erroneous value of zero used in many figures.

Figure 6 caption: typo "worth" -> "worse"

Corrected, thank you for catching it.

L303-310: reference specific panels of Figure 9 in the text to make the references more clear, in addition to panels a and b

Done.

Figure 9f: the pixel count of aspect drops to 0 at the far right – is this a clipping issue with the circular nature of aspect? I.e., 360 = 0? If so, recommend to remove the last value to avoid having a physical representation of the aspect distribution in this plot

Yes, this was the reason. We updated the figure accordingly, see updated Figure 9 below.

Figure 9: consider distinguishing the different lines more to make the figure more color vision deficiency and grayscale printing friendly, for example with dashed or dotted lines

We changed the grey line into dashed lines, see updated Figure 9 below. We decided to keep the black-green-red since it is used throughout the paper (i.e. also in Figures 10 and 11). However, we can clearly adapt it in a next iteration if needed, especially for colour blindness accessibility.

Figure 9: should be "number of pixels"

Indeed, this has been corrected, see updated Figure 9 below.

[Figure]

Updated version of Figure 9.

Figure 9: the terminology "training" when applied to datasets is confusing since that terminology is also applied to the training period within the framework.

This is the terminology used throughout the whole manuscript and introduced in P3L74-76. We would like to keep is unless considered seriously confusing.

Figure 10: there are many years represented here and with several regions and three lines per region, it is difficult to gain meaningful insight from this way of showing the data. Instead, consider a single DOY xaxis with transparent lines or a shaded region for each year and a solid line showing the different averages, or a scatter plot with training data on the x axis and model and corrected data in different shapes on the x axis with a 1-1 line, or some other setup

We agree, see the new version of Figure 10 above in our answer to the first reviewer.

L350-360: it is not entirely clear how the speedup factors in the text are related to the times shown in the table. For example, the 57-time speedup appears to compare 1-core R to 8-core C++. The 11x speedup in the correction phase from 1-core R to 1-core C++ is still impressive. It would be better to include data from all timings tested, for example showing a scaling plot with cores on the x axis and time on the y axis for the R and C++ implementation, and a second scaling plot with number of cores on the x axis and time on the y axis for both implementations.

There was indeed an error, thank you for catching it, and the correct value is 11. This has been updated in the revised Section 3.4.3 (see P20-21L402-413). We prefer to keep the data as a table as there are already 11 plots in the manuscript and we refer directly to numbers in the text, which are easier to read from a table.

L365: what is the time taken for disk access?

This was not directly monitored in our profiling. We only have the time spent in the C++ function reading the files and checking and storing the data in memory. As now explained in the updated version (see P21L414-420), this time is not pure disk access, which explains why there is still a benefit from parallelisation. We recognise that this is highly system dependent. We wish that this additional explanation answers your question about disk access.

L370: the results do seem to be sensitive to the parameters, and the authors make suggestions for optimal parameters in the paper

We updated the paragraph, see P21-22L429-431.

373: "remove" is an overstatement. Reduce.

It has been replaced by "reduce".

370-375: SnowQM as stated is used for bias correction, not error reduction. This should be consistent throughout the text. In particular, the authors state earlier in the paper and in the subsequent paragraph that the process does not remove the random error. Make sure this is consistent.

Thank you. We have replaced many instance of the word "error" with "bias" for consistency (when talking about mean error, bias is used, and "error" is used for other metrics).

403: reference to parallel computing is vague, specify the openMP approach

We simply removed the reference to parallel computing here since it is redundant with pervious text.

415: false positives?

No we mean false negative, i.e. days where the corrected model data has no snow when the training data has some.

L435 and abstract: abstract should not present repetitive results as the body of the text

We do not agree here. Numbers appearing in the abstract should appear somewhere in the body. However, here we have repetition between abstract, table 3, and conclusion. We removed the numbers from the conclusion.

**Reference**

Gutiérrez, J. M., A. S. Cofiño, R. Cano, and M. A. Rodríguez, 2004: Clustering Methods for Statistical Downscaling in Short-Range Weather Forecasts. Mon. Wea. Rev., 132, 2169–2183, https://doi.org/10.1175/1520-0493(2004)132<2169:CMFSDI>2.0.CO;2.

Fiddes, J. and Gruber, S.: TopoSCALE v.1.0: downscaling gridded climate data in complex terrain, Geosci. Model Dev., 7, 387–405, https://doi.org/10.5194/gmd-7-387-2014, 2014.

Maraun, D. Bias Correcting Climate Change Simulations - a Critical Review. Curr Clim Change Rep 2, 211–220 (2016). https://doi.org/10.1007/s40641-016-0050-x